# The CHORD protein CHP-1 regulates EGF receptor trafficking and signaling in *C. elegans* and in human cells

Andrea Haag[1,2†], Michael Walser[1†], Adrian Henggeler[1‡], Alex Hajnal[1*]

[1]Institute of Molecular Life Sciences, University of Zürich, Winterthurerstrasse, Switzerland; [2]Molecular Life Science Zürich PhD Program, Zürich, Switzerland

**Abstract** The intracellular trafficking of growth factor receptors determines the activity of their downstream signaling pathways. Here, we show that the putative HSP-90 co-chaperone CHP-1 acts as a regulator of EGFR trafficking in *C. elegans*. Loss of *chp-1* causes the retention of the EGFR in the ER and decreases MAPK signaling. CHP-1 is specifically required for EGFR trafficking, as the localization of other transmembrane receptors is unaltered in *chp-1(lf)* mutants, and the inhibition of *hsp-90* or other co-chaperones does not affect EGFR localization. The role of the CHP-1 homolog CHORDC1 during EGFR trafficking is conserved in human cells. Analogous to *C. elegans*, the response of CHORDC1-deficient A431 cells to EGF stimulation is attenuated, the EGFR accumulates in the ER and ERK2 activity decreases. Although CHP-1 has been proposed to act as a co-chaperone for HSP90, our data indicate that CHP-1 plays an HSP90-independent function in controlling EGFR trafficking through the ER.

*For correspondence:
alex.hajnal@imls.uzh.ch

[†]These authors contributed equally to this work

Present address: [‡]Institute of Biochemistry ETH Zürich, Zürich, Switzerland

Competing interests: The authors declare that no competing interests exist.

## Introduction

The generation and maintenance of cellular polarity is essential for the development and homeostasis of organs. Cell polarity governs various processes, such as cell migration, asymmetric cell division and morphogenesis (*Bryant and Mostov, 2008*). Most of these processes are regulated by extracellular signals, which are received and transduced by specific receptors on the plasma membrane. The intracellular trafficking and subcellular localization of these receptors in polarized epithelial cells profoundly affects their ligand-binding capabilities and the activation of the downstream signaling pathways. In particular, the EGFR family of receptor tyrosine kinases, which are activated by a multitude of ligands, play essential roles during the development of most epithelial organs (*Citri and Yarden, 2006*; *Sorkin and Goh, 2009*).

In contrast to mammals, *Caenorhabditis elegans* expresses only one EGFR homolog, LET-23, and a single EGF family ligand, LIN-3 (*Sundaram, 2006*). Thanks to this lack of redundancy, the *C. elegans* EGF/EGFR signaling system is well suited for systematic genetic analysis. LET-23 EGFR signaling controls a variety of developmental processes, including the development of the vulva, the egg-laying organ of the hermaphrodite (*Sternberg, 2005*). During vulval development, the six vulval precursor cells (VPCs) P3.p to P8.p are induced by an LIN-3 EGF signal from the anchor cell (AC) to differentiate into vulval cells (*Figure 1A*). The polarized distribution of LET-23 is crucial for the efficient activation of the downstream RAS/MAPK signaling pathway and the induction of the vulval cell fates (*Kaech et al., 1998*; *Whitfield et al., 1999*). P6.p, the VPC closest to the AC, receives the highest dose of LIN-3 and adopts the primary (1°) cell fate. At the same time, P6.p activates via a lateral Delta signal the LIN-12 Notch signaling pathway in its neighbors P5.p and P7.p, which inhibits the 1° and induces the secondary (2°) fate in these VPCs (*Sternberg, 2005*; *Berset et al., 2001*). The remaining VPCs P3.p, P4.p and P8.p that receive neither the inductive LIN-3 nor the lateral LIN-12 signal adopt the tertiary (3°) cell fate. The 3° VPCs divide once before fusing to the hypodermis

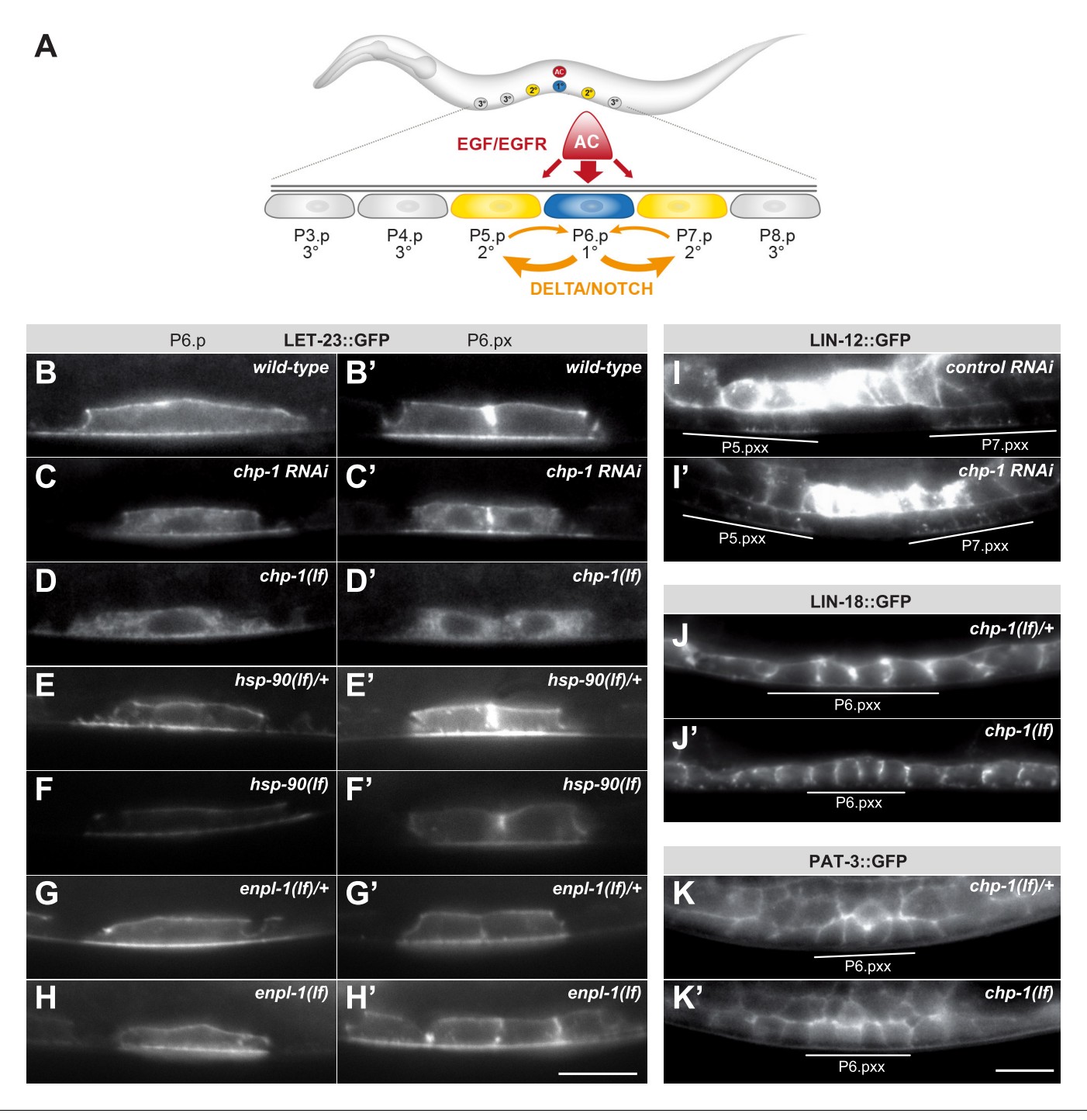

**Figure 1.** *chp-1* is required for the plasma membrane localization of LET-23::GFP. (**A**) Overview of the EGFR and NOTCH signaling pathways controlling VPC fate determination. (**B**) LET-23::GFP localization in P6.p and (**B'**) the two P6.p daughters (P6.px stage) of a wild-type, (**C, C'**) a *chp-1* RNAi and (**D, D'**) a homozygous *chp-1(tm2277lf)* mutant larva. (**E, E'**) LET-23::GFP expression in heterozygous *hsp-90(ok1333lf)/+* and (**F, F'**) and homozygous *hsp-90 (ok1333lf)* larvae at the Pn.p and Pn.px stage. (**G, G'**) LET-23::GFP expression in heterozygous *enpl-1(ok1964lf)/+* and (**H, H'**) homozygous *hsp-90 (ok1964lf)* larvae at the Pn.p and Pn.px stage. (**I**) LIN-12::GFP localization in a control RNAi and (**I'**) a *chp-1* RNAi-treated animals at the Pn.pxx stage. Note the unchanged apical localization of LIN-12::GFP in the 2° P5.p and P7.p descendants (underlined). (**J**) LIN-18::GFP membrane localization in a heterozygous *chp-1(tm2277lf)/+* and (**J'**) a homozygous *chp-1(tm2277lf)* mutant at the Pn.pxx stage. (K) PAT-3::GFP membrane localization in a heterozygous *chp-1(tm2277lf)/+* and (**K'**) a homozygous *chp-1(tm2277lf)* mutant at the Pn.pxx stage. The 1° P6.p descendants are underlined. At least 20 animals were analyzed for each genotype. The *chp-1*, *hsp-90* and *enpl-1* mutant phenotypes were completely penetrant, and *chp-1* RNAi perturbed LET-23 localization in more than 50% of the animals. The scale bars in (**H'**) and (**K'**) are 10 μm.

*Figure 1 continued on next page*

*Figure 1 continued*

The online version of this article includes the following figure supplement(s) for figure 1:

**Figure supplement 1.** LET-23::GFP localization in *hsp-90(p673)* mutants and after RNAi knock-down of co-chaperones.

hyp7. Hyperactivation of the EGFR/RAS/MAPK pathway causes more than three VPCs to adopt a vulval cell fate and a multivulva (Muv) phenotype, whereas reduced EGFR/RAS/MAPK signaling results in the induction of fewer than three VPCs and a vulvaless (Vul) phenotype.

Thanks to the availability of functional GFP tagged LET-23 reporters and the transparent body, vulval development is an excellent model to observe EGFR trafficking and localization in the epithelial VPCs of living animals (*Haag et al., 2014*). Before vulval induction, LET-23 is equally expressed in all VPCs. During induction, a positive MAP kinase MPK-1 feedback signal upregulates LET-23 expression in P6.p, which allows this cell to sequester most of the inductive LIN-3 EGF signal (*Stetak et al., 2006*). By contrast, LIN-12 NOTCH signaling in P5.p and P7.p results in the down-regulation of LET-23 and the inhibition of RAS/MAPK signaling (*Whitfield et al., 1999*).

Several factors that regulate RAS/MAPK activity by controlling the sub-cellular localization and trafficking of the LET-23 EGFR in the VPCs have been identified. The basolateral localization of LET-23 by the tripartite LIN-2/CASK, LIN-10/MINT and LIN-7/VELIS protein complex is necessary for efficient receptor activation, because LIN-3 is secreted by the AC in the somatic gonad facing the basolateral compartment of the VPCs (*Kaech et al., 1998*; *Hoskins et al., 1996*). On the other hand, the ARF GTPase exchange factor AGEF-1 antagonizes via the ARF GTPase and the AP-1 adaptor complex the basolateral localization of LET-23 (*Skorobogata et al., 2014*). A systematic in vivo screen in live *C. elegans* larvae has identified multiple additional regulators of LET-23 localization and signaling (*Haag et al., 2014*). One candidate identified in this screen was *chp-1*, which encodes a Cysteine and Histidine Rich Domain (CHORD) containing protein homologous to human CHORDC1 (also named Morgana) (*Brancaccio et al., 2003*; *Ferretti et al., 2011*). CHORDC1 has been proposed to function as co-chaperone with HSP90 (*Gano and Simon, 2010*), although CHORDC1 may also act independently of HPS90 in regulating microtubule dynamics (*Palumbo et al., 2020*).

Here, we show that a loss of *chp-1* function in *C. elegans* leads to the accumulation of LET-23 EGFR in the endoplasmic reticulum (ER) of the VPCs, resulting in a strongly reduced activity of the RAS/MAPK pathway. CHP-1 is specifically required for LET-23 localization, as the secretion of other transmembrane receptors to the VPC plasma membrane is unchanged in the absence of *chp-1*. Furthermore, we shown that deletion of CHORDC1 in human A431 cells leads to the ER mislocalization of the EGFR and to a loss of EGF-induced filopodia formation. Analogous to *C. elegans chp-1*, deletion of human CHORDC1 does not eliminate but rather attenuates the activation of the MAPK pathway in response to EGF stimulation. We propose that CHP-1 CHORDC1 plays a conserved and specific function during the maturation and membrane secretion of the EGFR.

## Results

### *chp-1* is required for basolateral localization of the EGFR LET-23

The basolateral localization of the LET-23 EGFR in the VPCs of *C. elegans* larvae is necessary for the efficient binding of the LIN-3 EGF ligand secreted by the AC (*Figure 1A*; *Kaech et al., 1998*; *Whitfield et al., 1999*). After ligand-induced receptor endocytosis from the basolateral plasma membrane, LET-23 accumulates on the apical cortex of the VPCs (*Haag et al., 2014*). In late L2/early L3 larvae before the VPCs have started dividing, a translational LET-23::GFP reporter was up-regulated in the 1° VPC P6.p, while expression faded in the other, more distal VPCs. Most of the LET-23::GFP protein in P6.p was detected, approximately at equal levels, on the basolateral and apical plasma membranes (*Figure 1B*). Only a faint and diffuse intracellular LET-23::GFP signal could be observed in the VPCs of wild-type animals. After the first round of VPC divisions, LET-23::GFP continued to be strongly expressed on the plasma membrane of the 1° P6.p descendants (*Figure 1B'*). A systematic RNA interference screen for genes controlling LET-23 trafficking had previously identified the *chp-1* gene as a regulator of LET-23 localization (*Haag et al., 2014*). *chp-1* encodes a conserved CHORD-containing protein homologous to human CHORDC1/Morgana (*Ferretti et al., 2011*). RNAi against *chp-1* leads to a strong reduction in plasma membrane localization and to the

intracellular accumulation of the LET-23::GFP reporter in P6.p and its descendants (*Figure 1C,C'*). To confirm the RNAi-induced phenotype, we examined LET-23 localization in *chp-1(tm2277)* deletion mutants (*chp-1(lf)*). Since homozygous *chp-1(lf)* mutants are sterile as adults, we analyzed LET-23 EGFR localization in the homozygous offspring of heterozygous *chp-1(lf)/hT2* balanced mothers (*Supplementary file 1*). In the following experiments, we compared *chp-1(lf)* homozygous larvae to balanced *chp-1(lf)/hT2* heterozygous control siblings, since LET-23::GFP localization in *chp-1(lf)/+* heterozygotes was indistinguishable from wild-type animals (for example, compare *Figure 1B* with Figure 3A). Homozygous *chp-1(lf)* L2 and L3 larvae exhibited a completely penetrant intracellular mislocalization of LET-23 EGFR in the VPCs and their descendants (*Figure 1D,D'*). The distinct plasma membrane signal observed in the VPCs of wild-type animals was absent in *chp-1(lf)* larvae.

Taken together, the CHORD-containing protein CHP-1 is required for the plasma membrane localization of the EGFR in the VPCs.

## *chp-1* acts independently of the *hsp-90A* and *hsp-90B1* chaperones

In mammalian cells, CHORDC1/Morgana forms a complex with the heat-shock protein 90 (HSP90) and was proposed to function as a co-chaperone for a subset of the HSP90 clients (*Gano and Simon, 2010*). We therefore tested if a loss-of-function mutation in the *C. elegans hsp-90* gene affects LET-23 EGFR localization in as similar manner as *chp-1(lf)*. The *hsp-90(ok1333)* allele contains a 1258 bp deletion resulting in the insertion of a Ser followed by a stop codon after amino acid 208, thereby truncating the remaining 494 residues of the protein (www.wormbase.org). The *ok1333* allele was previously reported to eliminate *hsp-90* function (*Lissemore et al., 2018*). Homozygous, *hsp-90 (ok1333)* larvae segregated by heterozygous *nT1* balanced mothers (*Supplementary file 1*) displayed a paralyzed Unc phenotype and arrested at the mid to late L3 stage. We thus examined LET-23::GFP expression in the VPCs of homozygous *hsp-90(ok1333)* larvae shortly before they arrested, at the late L2 to early L3 stage (Pn.p to Pn.px stage). While the overall intensity of the LET-23::GFP signal in the VPCs of homozygous *hsp-90(ok1333)* mutants was reduced compared with their heterozygous siblings, the LET-23::GFP signal remained localized at the plasma membrane (*Figure 1E–F'*). Especially, we did not observe the intracellular LET-23::GFP accumulation seen in *chp-1(lf)* mutants (*Figure 1D*). A similar reduction in LET-23::GFP expression with persisting membrane localization was observed in larvae carrying the *hsp-90(p673)* missense mutation (*Figure 1—figure supplement 1A,A'*). Even though the *p673* allele acts as a gain-of-function allele with respect to the constitutive Dauer (Daf-c) phenotype (*Birnby et al., 2000*), our data suggest that for LET-23 membrane localization *p673* rather represents a reduction-of-function allele.

The *C. elegans* genome encodes an *hsp-90* paralog, called *enpl-1*, which is homologous to the human HSP90B1 chaperone (also called GP96 or GRP94) (*Natarajan et al., 2013*). Since mammalian HSP90B1 controls the trafficking of Toll-like receptors through the endoplasmic reticulum (ER) (*Randow and Seed, 2001*), we tested whether ENPL-1 instead of HSP-90 might act together with CHP-1 to regulate LET-23 localization. For this purpose, we used a 1613 bp *enpl-1* deletion allele *ok1964* that removes most of the coding sequences and eliminates gene function (*Natarajan et al., 2013*). Homozygous *enpl-1(ok1964)* larvae did not exhibit a change in LET-23::GFP expression or localization in the VPCs (*Figure 1G–H'*), indicating that ENPL-1 is not required for LET-23 membrane localization. Finally, we performed RNAi knock-down of other known HSP90 co-chaperones, *cdc-37*, *daf-41* and *sgt-1* (*Li et al., 2012*), but observed no change in LET-23::GFP localization (*Figure 1—figure supplement 1B–F'*).

We conclude that CHP-1 acts independently of the HSP90A and HSP90B1 chaperones to control the plasma membrane localization of the LET-23 EGFR in the VPCs.

## *chp-1* is a specific regulator of LET-23 localization in the VPCs

To investigate if *chp-1* plays a general role in membrane trafficking, we examined the expression pattern of three other transmembrane receptors expressed in the VPCs; LIN-12 NOTCH (*Shaye and Greenwald, 2002*), LIN-18 RYK (*Inoue et al., 2004*) and the β-integrin subunit PAT-3 (*Hagedorn et al., 2009*). The translational LIN-12::GFP reporter was expressed on the apical membrane of the VPCs (*Figure 1G*), while the LIN-18::GFP and PAT-3::GFP translational reporters localized predominantly to the basolateral compartment (*Figure 1H,I*). *chp-1* RNAi did not alter the

apical localization of the LIN-12::GFP reporter (*Figure 1G'*). Furthermore, the *chp-1(lf)* mutation did not affect the basolateral localization of the LIN-18::GFP or the PAT-3::GFP reporter (*Figure 1H',I'*).

Thus, CHP-1 does not play a general role in the apical or basolateral secretion of transmembrane receptors, but it is rather specifically required for the membrane localization of the EGFR.

## *chp-1* acts cell autonomously in the VPCs

We next tested if *chp-1* acts cell-autonomously in the VPCs. First, we performed tissue-specific RNAi to downregulate *chp-1* expression in the VPCs. For this purpose, we used a transgenic line expressing the *rde-1(+)* gene under control of the Pn.p cell-specific promoter/enhancer in an *rde-1(lf)* RNAi-resistant background carrying the *let-23::gfp* reporter (*Haag et al., 2014*) (strain AH2417 in *Supplementary file 1*). This approach has previously been shown to downregulate gene expression in a tissue-specific manner (*Qadota et al., 2007*). RNAi against *chp-1* in the Pn.p cell-specific RNAi strain caused an intracellular accumulation of LET-23::GFP in 31% of the cases, while control animals treated with the empty RNAi vector showed no change in LET-23::GFP localization (*Figure 2A–B'*).

In addition, we used a tissue-specific CRISPR/Cas9 approach to inactivate *chp-1* in the VPCs (*Shen et al., 2014*). For this purpose, we expressed the Cas9 endonuclease under control of a 1° VPC-specific *egl-17* promoter fragment (*Inoue et al., 2002*) together with two single-guide (sg) RNAs that target the first exon of *chp-1* and were expressed under control of the ubiquitous *eft-3* promoter (array *zhEx558* in strain AH4609 in *Supplementary file 1*). In around 5% of *zhEx558* animals, we observed an intracellular accumulation of LET-23::GFP (*Figure 2D,D'*). In all these cases, LET-23::GFP was mislocalized only in the 1° VPC P6.p and its descendants. Control sibling lacking the *zhEx558* array showed a wild-type membrane localization of LET-23::GFP (*Figure 2C,C'*). The

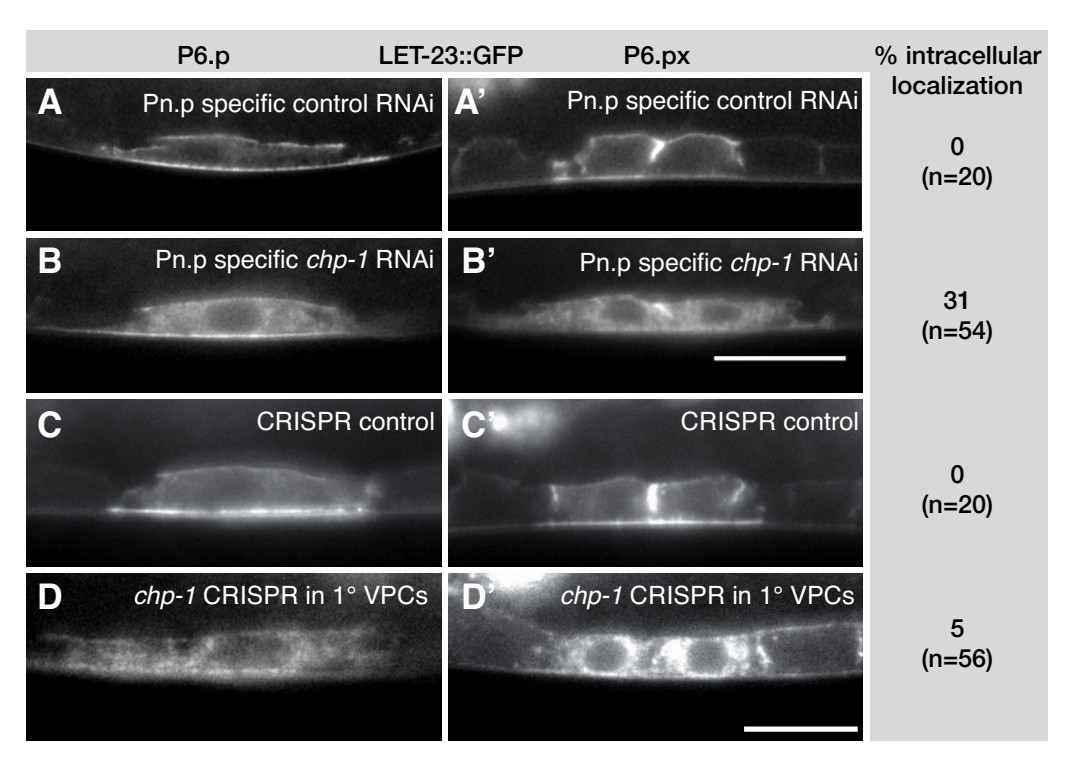

**Figure 2.** *chp-1* acts cell-autonomously in the VPCs. (A, A') LET-23::GFP localization in the VPCs of empty vector control and (B, B') *chp-1* RNAi treated animals in the Pn.p cell-specific RNAi background. (C, C') LET-23::GFP expression in a control siblings without and (D, D') with the *zhEx558[chp-1sg, egl-17p::cas-9]* transgene. Note in (D') the intracellular mislocalization of LET-23::GFP in the two P6.p descendants, while LET-23::GFP remained localized at the plasma membrane in the adjacent P7.p descendant. For each condition, the frequencies of the LET-23::GFP mislocalization phenotype and the numbers of animals analyzed are indicated to the right. The scale bars in (B') and (D') are 10 µm.

relatively low penetrance of the CRISPR/CAS9-induced mislocalization phenotype compared to the tissue-specific RNAi could be due to an inefficient binding of the sgRNAs to the target sequence, to mosaic expression of the extrachromosomal array or to the perdurance of the CHP-1 protein in the VPCs.

Together, these experiments indicated that CHP-1 acts cell-autonomously to regulate LET-23::GFP localization in the VPCs.

## Loss of *chp-1* function causes an accumulation of LET-23 EGFR in the endoplasmic reticulum

The intracellular accumulation of LET-23::GFP in *chp-1(lf)* mutants appeared granular and unevenly structured, while in *chp-1(lf)/+* control animals, LET-23::GFP was localized predominantly at the plasma membrane of the 1° VPC P6.p and its descendants (*Figure 3A–C*; note that single mid-sagittal confocal sections through the VPCs are shown in *Figure 3*, whereas *Figures 1*, *2* and *4* shows wide-field images of the entire VPCs.) In order to determine the intracellular compartment, in which LET-23::GFP accumulates in *chp-1(lf)* mutants, we generated two reporters that mark the Golgi and the ER of the VPCs, respectively. To label the Golgi apparatus, we expressed an alpha-mannosidase 2A AMAN-2::mCherry fusion protein in the VPCs under control of the pan-epithelial *dlg-1* promoter. AMAN-2 has previously been shown to localize to the Golgi network in the *C. elegans* intestine (*Chen et al., 2006*). In contrast to vertebrate cells that contain one large juxta-nuclear Golgi stack, invertebrate cells contain many small Golgi stacks (Golgi 'ministacks') dispersed throughout the

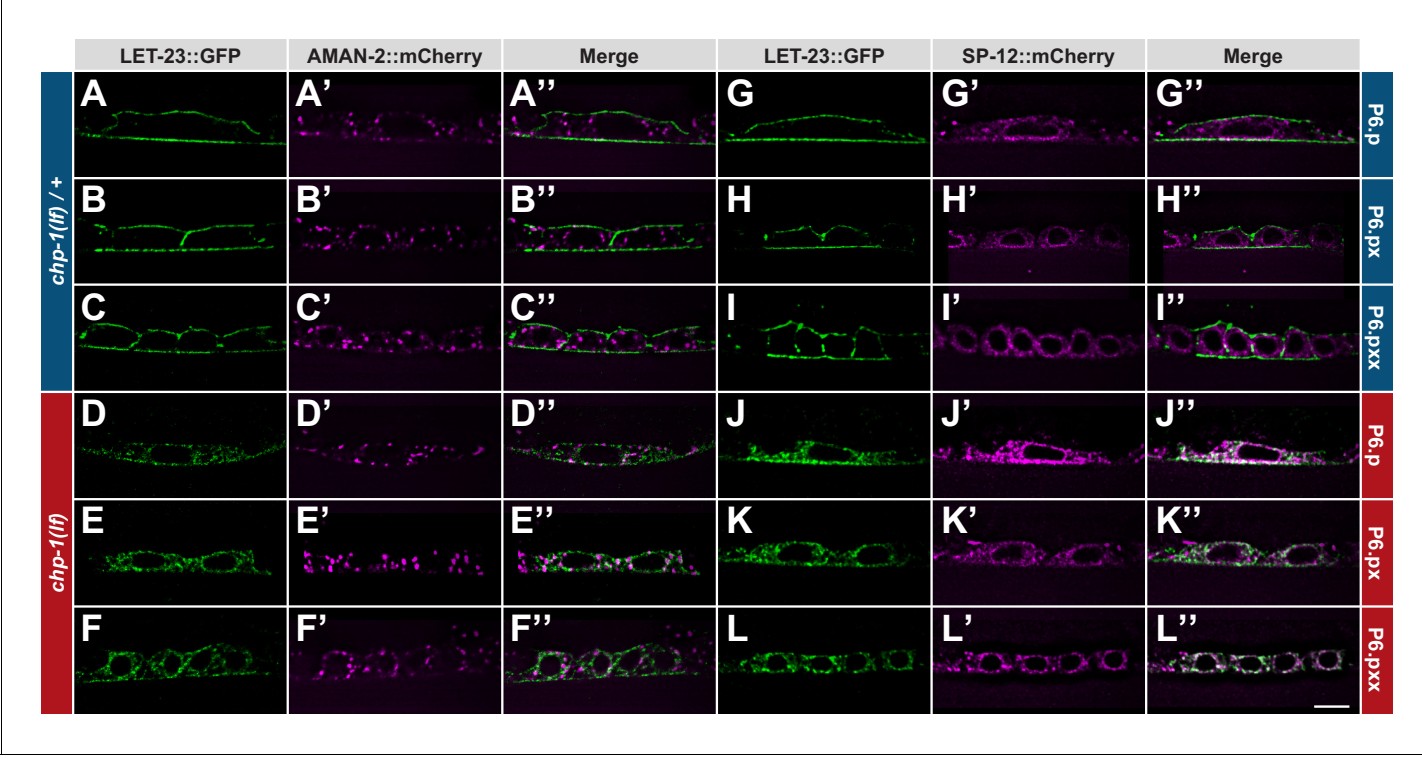

**Figure 3.** ER mislocalization of LET-23::GFP in *chp-1(lf)* mutants. (**A–C''**) Localization of LET-23::GFP and the AMAN-2::mCherry Golgi marker in heterozygous *chp-1(tm2277lf)/+* control siblings and (**D–F''**) homozygous *chp-1(tm 2277lf)* mutants at the P6.p to P6.pxx stage. (**G–I''**) Localization of LET-23::GFP and the SP12::mCherry ER marker in heterozygous *chp-1(tm2277lf)/+* control siblings and (**J–L''**) homozygous *chp-1(tm 2277lf)* mutants at the P6.p to P6.pxx stage. The individual panels show the different channels of single mid-sagittal confocal sections through P6.p or its descendants. A voxel by voxel quantification of the co-localization between the AMAN-2::mCherry (Golgi) and SP12::mCherry (ER) markers with the LET-23::GFP signal is shown in *Figure 3—figure supplement 1*. The scale bar in (**L''**) is 10 µm.

The online version of this article includes the following figure supplement(s) for figure 3:

**Figure supplement 1.** Quantification of the co-localization between the AMAN-2::mCherry (Golgi) and SP12::mCherry (ER) markers with LET-23::GFP in P6.p and its descendants.

**Figure supplement 2.** Unfolded protein response in *chp-1(lf)* mutants after tunicamycin treatment.

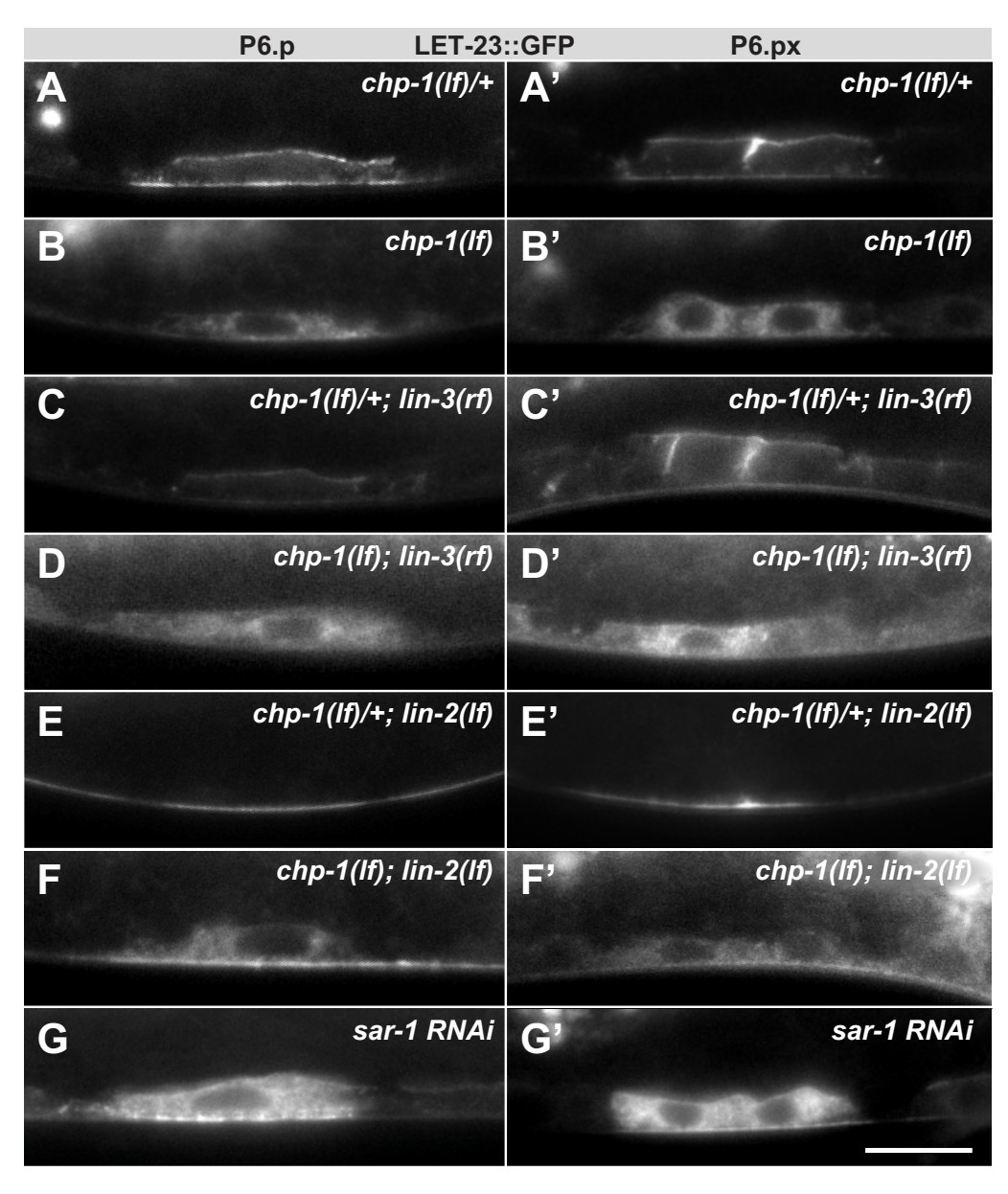

**Figure 4.** Intracellular mislocalization of LET-23::GFP in *chp-1(lf)* mutants is ligand independent. (**A, A'**) Localization of LET-23::GFP in heterozygous *chp-1(tm2277lf)/+* control siblings, (**B, B'**) homozygous *chp-1(tm 2277lf)* mutants, (**C, C'**) *chp-1(tm2277lf)/+; lin-3(e1417rf)* heterozygous, and (**D, D'**) in *chp-1(tm2277lf); lin-3(e1417rf)* homozygous double mutants at the P6.p and P6.px stage. (**E, E'**) Apical mislocalization of LET-23::GFP in *chp-1(tm2277lf)/+; lin-2(n397lf)* heterozygous and (**F, F'**) in *chp-1(tm2277lf); lin-2(n397lf)* homozygous double mutants. At least 20 animals were analyzed for each condition, and animals of the same genotype all showed the same LET-23::GFP localization pattern. (**G, G'**) *sar-1* RNAi causes the same intracellular accumulation of LET-23::GFP as *chp-1(lf)*. The scale bar in (**G'**) is 10 μm.

cytoplasm (*Ripoche et al., 1994*). Accordingly, the AMAN-2::mCherry reporter labeled punctate structures scattered throughout the cytoplasm of the VPCs (*Figure 3A'–F'*). In *chp-1(lf)/+* animals, LET-23::GFP showed on average 26% co-localization with the AMAN-2::mCherry reporter when analyzed on a voxel per voxel basis in confocal optical sections of the VPCs (*Figure 3A–C''* and *Figure 3—figure supplement 1*). In homozygous *chp-1(lf)* mutants, the co-localization with AMAN-2::mCherry was slightly increased to 34.4% (*Figure 3D–F''* and *Figure 3—figure supplement 1*).

However, the strongest LET-23::GFP signal was detected at the plasma membrane, indicating that only a minor fraction of LET-23::GFP is found in the Golgi apparatus.

To label the ER compartment of the VPCs, we created a reporter consisting of the *dlg-1* promoter driving expression of an *mCherry* tag C-terminally fused to the C34B2.10 gene, which encodes the 12 kDa subunit (SP12) of the *C. elegans* ER signal peptidase complex (*Rolls et al., 2002*). Translational SP12 reporters have previously been shown to localize to a reticular tubular network that extends to the cortex in various cell types of *C. elegans* and resembles the ER architecture in yeast and mammalian cells (*Voeltz et al., 2002*). In confocal sections through the VPCs of *chp-1(lf)/+* animals, we observed 34% co-localization between the SP12::mCherry and LET-23::GFP reporters, though the strongest LET-23::GFP signal intensity was found at the cell cortex where no SP12::mCherry was detected (*Figure 3G–I''* and *Figure 3—figure supplement 1*). By contrast, in homozygous *chp-1(lf)* mutants, we observed a strong overlap between the LET-23::GFP and SP12::mCherry signals inside the cells, resulting in 64% co-localization between the two reporters (*Figure 3J–L''* and *Figure 3—figure supplement 1*).

Therefore, the loss of *chp-1* function leads to the intracellular retention of LET-23 predominantly in the ER.

## ER mislocalization of LET-23 in *chp-1(lf)* mutants does not activate the unfolded protein response pathway

Since LET-23::GFP accumulated mainly in the ER compartment of *chp-1(lf)* mutants, we tested if loss of *chp-1* function causes ER stress triggering the unfolded protein response (UPR) pathway. The *hsp-4* gene encodes a homolog of the mammalian Grp78/BiP protein that is upregulated upon ER stress via the XBP-1 transcription factor and the IRE-1 kinase/endoribonuclease (*Calfon et al., 2002*). The expression of an *hsp-4::gfp* reporter thus serves as an in vivo readout for the UPR (*Taylor and Dillin, 2013*). As a positive control, we treated animals with tunicamycin, which induces UPR by blocking the formation of N-acetylglucosamine lipid intermediates necessary for the glycosylation of newly synthesized proteins in the ER (*Taylor and Dillin, 2013*). Untreated *chp-1(lf)* mutants did not show elevated *hsp-4*::GFP expression when compared to the wild-type (*Figure 3—figure supplement 2A, C,E*). A 4 hr exposure of young adult wild-type animals to 25 μg/ml tunicamycin caused an approximately eightfold increase in *hsp-4*::GFP fluorescence intensity (*Figure 3—figure supplement 2B,E*). Interestingly, tunicamycin-treated *chp-1(lf)* mutants exhibited a stronger induction of *hsp-4*::GFP expression (*Figure 3—figure supplement 2D,E*).

Taken together, the intracellular accumulation of LET-23 in the *chp-1(lf)* mutants does not activate the UPR pathway under standard conditions. However, *chp-1(lf)* mutants are slightly hypersensitive to ER stress induced by tunicamycin-treatment.

## Intracellular LET-23 EGFR accumulation in *chp-1(lf)* mutants is ligand-independent

Binding of the LIN-3 EGF ligand to the LET-23 EGFR on the basolateral cortex of the VPCs induces rapid receptor endocytosis (*Haag et al., 2014*). The endocytosed LET-23 may be recycled to the basolateral compartment, accumulate in the apical membrane compartment or undergo lysosomal degradation (*Stetak et al., 2006*; *Skorobogata and Rocheleau, 2012*). In *lin-3(e1417)* mutants that lack LIN-3 expression in the AC (*Hwang and Sternberg, 2004*), LET-23::GFP accumulated on the basolateral cortex of the VPCs, while the apical LET-23::GFP signal was reduced (*Figure 4C*), as reported previously (*Haag et al., 2014*). We thus examined whether the intracellular accumulation of LET-23 in *chp-1(lf)* mutants depends on ligand-induced receptor endocytosis. In *lin-3(e1417); chp-1 (lf)* double mutants, we observed the same intracellular accumulation of LET-23::GFP as in *chp-1(lf)* single mutants (*Figure 4B,D*). Thus the intracellular accumulation of LET-23::GFP in *chp-1(lf)* mutants does not depend on ligand-induced receptor endocytosis.

Next, we asked whether *chp-1* acts at the level of the tripartite LIN-2/LIN-7/LIN-10 complex, which is required for the basolateral retention of LET-23 and facilitates ligand binding (*Kaech et al., 1998*; *Whitfield et al., 1999*). Mutations in *lin-2*, *lin-7* or *lin-10* cause a penetrant vulvaless (Vul) phenotype because LET-23 is mislocalized to the apical membrane compartment, where it cannot bind to LIN-3 secreted by the AC on the basal side of the VPCs (*Simske et al., 1996*; *Whitfield et al., 1999*). In *lin-2(lf)* single mutants, the LET-23::GFP signal was detected almost exclusively on the

apical membranes of the VPCs (*Figure 4E*). In *chp-1(lf)*; *lin-2(lf)* double mutants, most of the LET-23:: GFP signal was found in the intracellular compartment similar to *chp-1(lf)* single mutants (*Figure 4F*). However, we did observe a faint LET-23::GFP signal on the apical cortex in *chp-1(lf)*; *lin-2(lf)* double mutants, indicating that a small fraction of LET-23::GFP can be secreted to the apical plasma membrane in the absence of *chp-1*. Finally, RNAi against *sar-1*, which encodes a small GTP-binding protein required for ER to Golgi transport, caused the same intracellular mislocalization of LET-23::GFP as observed in *chp-1(lf)* mutants (*Figure 4G*).

We thus conclude that CHP-1 neither regulates the ligand-induced endocytosis nor the basolateral retention of LET-23, but rather controls the secretion of the receptor from the ER to the plasma membrane.

## *chp-1* is a positive regulator of EGFR/RAS/MAPK signaling in the VPCs

The basolateral membrane localization of LET-23 is necessary for efficient ligand binding and activation of the downstream RAS/MAPK signaling pathway in the VPCs (*Kaech et al., 1998*; *Hoskins et al., 1996*; *Whitfield et al., 1999*). To quantify the output of the RAS/MAPK pathway in the VPCs, we quantified the expression of a transcriptional P$_{egl-17}$::*cfp* reporter as a marker for the 1° cell fate (*Burdine et al., 1998*). *egl-17* encodes a fibroblast growth factor (FGF)-like protein, which is upregulated by RAS/MAPK signaling in the 1° VPC (P6.p) and its descendants until the late L3 stage. P$_{egl-17}$::*cfp* expression levels in *chp-1(lf)* mutants at the Pn.pxx stage were decreased around five-fold when compared to wild-type larvae at the same stage (*Figure 5A,B,E*). We further examined the activity of the lateral NOTCH signaling pathway using a transcriptional P$_{lip-1}$::*gfp* reporter that is upregulated in 2° VPC in response to LIN-12 NOTCH activation (*Berset et al., 2001*). Expression of the P$_{lip-1}$::*gfp* reporter was unchanged in *chp-1(lf)* mutants (*Figure 5C,D*). Thus, CHP-1 acts as a positive regulator of RAS/MAPK signaling, while the activity of the lateral NOTCH pathway is not affected by *chp-1(lf)*.

Despite the strong reduction in RAS/MAPK reporter expression, the VPCs in *chp-1(lf)* mutants were induced to proliferate and differentiate into vulval cells. In most *chp-1(lf)* single mutants, the three proximal VPCs P5.p, P6.p and P7.p differentiated as in wild-type animals (*Figure 4F,G*). However, the vulval invagination formed by the descendants of the three induced VPCs had an abnormal shape and the vulval cells formed two separate invaginations, indicating that CHP-1 performs additional functions during vulval fate execution or morphogenesis (*Figure 5G*). To quantify vulval induction, we determined the vulval induction index (VI) by counting the average number of VPCs per animals that were induced to differentiate, as described (*Schmid et al., 2015*). A VI of 3 indicates wild-type differentiation, while a VI > 3 signifies over- and a VI < 3 under-induction. The VI thus serves as a quantitative readout to examine genetic interactions between signaling pathway components.

Most *chp-1(lf)* mutants showed a VI of 3, though we observed over- as well as under-induced animals (*Figure 5L*). To investigate the interaction between *chp-1* and the EGFR/RAS/MAPK signaling pathway, we constructed double mutants between *chp-1(lf)* and core EGFR/RAS/MAPK pathway components. The *lin-3(e1417rf)* allele caused a strong reduction in the VI of *chp-1(lf)* mutants, approximately to the level of *lin-3(e1417)* single mutants (*Figure 5H,I,L*; *Hwang and Sternberg, 2004*). This indicated that the VPCs in *chp-1(lf)* mutants are at least partially sensitive to the inductive AC signal. The *let-23(sy1)* allele specifically prevents the interaction of LET-23 EGFR with the LIN-2/ LIN-7/LIN-10 receptor localization complex and causes a similar apical receptor mislocalization and partially penetrant Vul phenotype as the *lin-2(lf)* mutation (*Kaech et al., 1998*; *Whitfield et al., 1999*). Double mutants between *chp-1(lf)* and *let-23 egfr(sy1)* or *lin-2(lf)* exhibited a significantly stronger Vul phenotype than either of the single mutants (*Figure 5L*). Thus, *chp-1(lf)* enhanced the Vul phenotype caused by apical mislocalization of LET-23 EGFR. By contrast, the Muv phenotype caused by the *n1046* gain-of-function (*gf*) mutation in the *let-60 ras* gene (*Beitel et al., 1990*) was significantly enhanced by *chp-1(lf)* (*Figure 5J–L*). Similar to the apical mislocalization in *lin-2(lf)* mutants, the intracellular mislocalization of LET-23 in *chp-1(lf)* mutants likely results in decreased ligand sequestering by P6.p, which allows more LIN-3 signal to diffuse to the distal VPCs (*Hajnal et al., 1997*). In combination with the hyper-sensitive *let-60(n1046gf)* background, reduced ligand sequestering results in the induction of the distal VPCs and a hyperinduced phenotype (*Hajnal et al., 1997*).

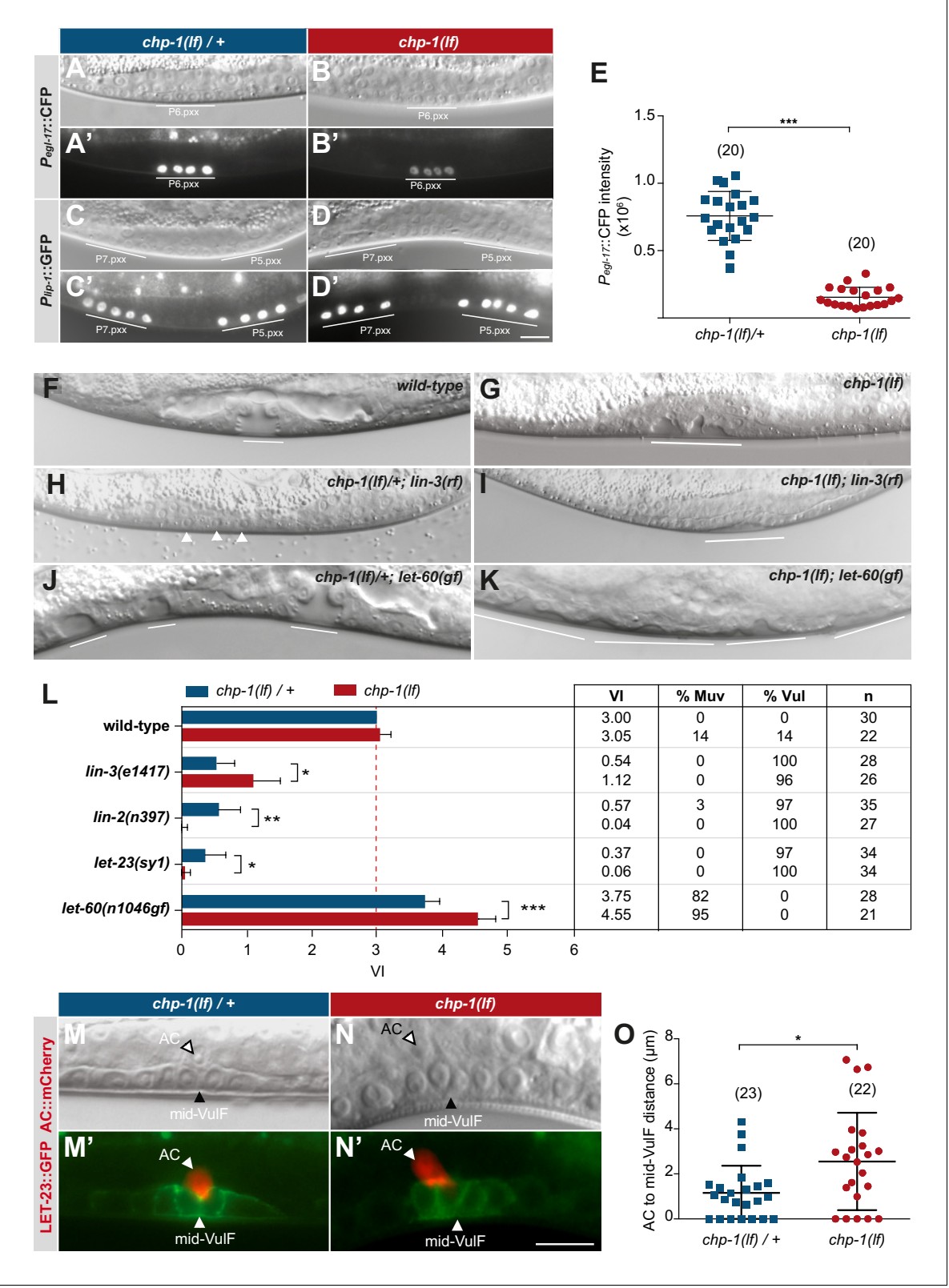

**Figure 5.** CHP-1 positively regulates EGFR/RAS/MAPK signaling in the VPCs. (A, A') Expression of the 1° cell fate reporter P*egl-17*::CFP in heterozygous *chp-1(tm2277lf)/+* and (B, B') homozygous *chp-1(lf)* mutant at the Pn.pxx stage. The top panels show Nomarski images of the differentiating VPCs and the bottom panel the reporter expression taken with identical exposure settings. (C, C') Expression of the 2° cell fate reporter P*lip-1*::GFP in heterozygous *chp-1(tm2277lf)/+* and (D, D') homozygous *chp-1(lf)* mutant at the Pn.pxx stage. The scale bar in (D') is 10 µm. (E) Quantification of the

*Figure 5 continued on next page*

*Figure 5 continued*

P*egl-17*::CFP fluorescence intensity in the 1° VPCs at the Pn.pxx stage. The p-value was calculated by a two-tailed t-test for independent samples. The numbers of animals quantified are indicated in brackets. (F) Nomarski images of the vulval morphology in wild-type, (G) homozygous *chp-1(tm2277lf)*, (H) heterozygous *chp-1(tm2277lf)/+; lin-3(e1417rf)*, (I) homozygous *chp-1(tm2277rf); lin-3(e1417rf)*, (J) heterozygous *chp-1(tm2277lf)/+; let-60(n1046gf)* and (K) homozygous *chp-1(tm2277lf); let-60(n1046gf)* L4 larvae. The descendants of induced VPCs forming an invagination are underlined and the arrowheads in (H) point at the nuclei of uninduced VPCs. (L) Quantification of the vulval induction index (VI) for the indicated genotypes. The table to the right shows the absolute mean VI, the percentage of animals with a Muv (VI > 3) and a Vul (VI < 3) phenotype and the number of animals scored (n) for each genotype. Error bars indicating the 95% confidence intervals and p-values were calculated by Bootstrapping with a resampling size of 1000, as described in *Maxeiner et al. (2019)* (p<0.05 = *p<0.01 = ** and p<0.001 = ***). (M, M') AC to 1° VPC alignment at the Pn.pxx stage in heterozygous *chp-1(tm2277lf)/+* and (N, N') homozygous *chp-1(tm2277lf)* mutants. The top panels show Nomarski images and the bottom panels the expression of the LET-23::GFP reporter in green and the *qyIs23[Pcdh-3:: PLC∂PH::mCherry]* reporter labeling the AC in red. The scale bar in (N') is 10 µm. (O) Quantification of the AC to VulF midline distance. The p-value was calculated by a two-tailed t-test for independent samples, and the numbers of animals scored are indicated in brackets.

In summary, our genetic analysis indicated that *chp-1* positively regulates EGFR/RAS/MAPK signaling in the VPCs. However, the VPCs in *chp-1(lf)* mutants are capable of differentiating into vulval cells because they remain partially sensitive to the inductive LIN-3 EGF signal.

## CHP-1 is necessary for the precise AC to P6.p alignment

Besides VPC fate specification, LIN-3 to LET-23 signaling is also required for the proper alignment between the AC and the 1° VPC P6.p (*Grimbert et al., 2016*). In wild-type L2 stage larvae, the relative position between the AC and VPCs is highly variable. However, by the early L3 stage the 1° VPC P6.p has migrated toward the AC such that the AC and P6.p are precisely aligned with each other. In wild-type larvae at the mid-L3 (Pn.pxx) stage, the AC was located at the vulval midline above the two inner 1° P6.p descendants (the VulF cells) (*Figure 5M,M'*). By contrast, in *chp-1(lf)* mutants the AC was often misplaced and occasionally located between VulF and VulE (*Figure 5N,N*). To quantify the AC to VulF alignment, we measured the distance between the AC nucleus and the midpoint between the two VulF cells at the Pn.pxx stage. In *chp-1(lf)/+* heterozygous control animals, the AC to mid-VulF distance was around 1 µm or smaller, while most *chp-1(lf)* mutants exhibited a distance greater than 1 µm (*Figure 5O*).

Thus, in addition to VPC fate specification *chp-1* is also required for the precise alignment between the AC and the VPCs, which is also mediated by LIN-3/LET-23 signaling.

## Human CHORDC1 is required for filopodia formation and sustained ERK1/2 activation by the EGFR

To examine if the role of CHP-1 in controlling EGFR localization and signaling is conserved in mammalian cells, we generated a CRISPR/Cas9-mediated knock-out of the mammalian *chp-1* homolog CHORDC1 in cultured cells. We used the human vulva epidermoid carcinoma cell line A431 because these cells express high levels of wild-type EGFR and respond strongly to EGF stimulation (*Van de Vijver et al., 1991*). A431 cells were transduced with lentiviral particles that deliver Cas9 and two sgRNAs targeting the first exon of CHORDC1 (see Materials and methods). As negative control, cells were transduced with a lentivirus delivering a scrambled sgRNA. After bulk puromycin selection to eliminate uninfected cells, two-cell populations were generated, subsequently termed A431 KO and A431 control cells, respectively. Western blot analysis revealed a 96% reduction of CHORDC1 protein levels in A431 KO cells (*Figure 6A*).

A431 KO cells displayed a reduced growth rate and a tendency to grow in smaller, scattered patches when compared to A431 control cells (*Figure 6B,C*). Furthermore, loss of CHORDC1 resulted in cell lethality approximately 16 days post lentiviral transduction, which made it impossible to establish A431 KO lines from single-cell clones. Therefore, the following experiments were performed with populations of A431 KO and control cells 10 to 14 days post-lentiviral transduction. To further characterize the morphological defects of A431 KO cells, we visualized the actin cytoskeleton by phalloidin staining. The numerous F-actin rich filopodia protruding from the plasma membrane of A431 control cells were absent in A431 KO cells (*Figure 6D,E*). Instead, A431 KO cells contained densely packed cortical F-actin filaments arranged in a circumferential manner. A similar phenotypic

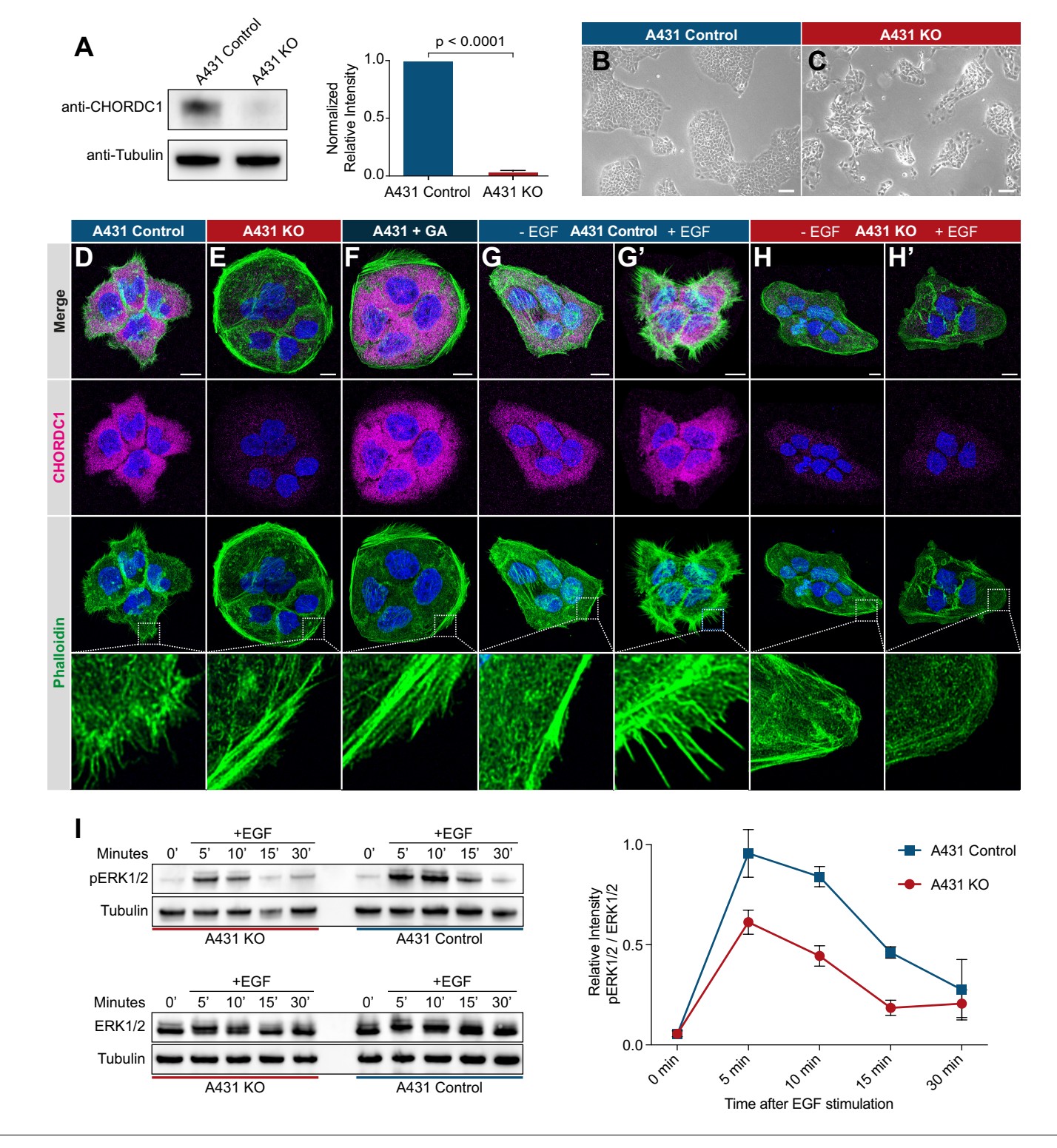

**Figure 6.** CHORDC1 is required for EGF-induced filopodia formation and sustained ERK activation in A431 cells. (**A**) Quantification of CHORDC1 protein levels in A431 control and KO cells by Western blot analysis (average of eleven biological replicates). The bar graph shows the normalized averaged relative intensities ± SEM. The p-value was calculated using a two-tailed t-test for independent samples. (**B**) Phase contrast images of A431 control and (**C**) A431 KO cells 12 days post-lentiviral transduction. The scale bars are 100 µm. (**D**) Immunofluorescence staining of A431 control cells with antibodies recognizing CHORDC1 (magenta), fluorescently labeled phalloidin (green) and DAPI (blue), (**E**) A431 KO cells, and (**F**) A431 cells treated for 24 hr with 1 µM geldanamycin (GA). (**G, H**) Control- and KO cells fixed after 16 hr of serum starvation, and (**G', H'**) 10 min after stimulation with 100

*Figure 6 continued on next page*

*Figure 6 continued*

ng/ml human EGF. The scale bars are 10 μm. The bottom row shows higher magnifications of the cortical regions outlined by the dashed squares. (I) Total protein lysates of serum-starved cells that had been stimulated with 100 ng/ml human EGF for the indicated times (in minutes) were analyzed on Western blots with antibodies against phospho-ERK1/2 and total ERK1/2. The graph to the right shows the relative phospho-ERK1/2 signals normalized to the total ERK1/2 levels at each time point. The data shown represent the average ratios obtained in three biological replicates. Error bars indicate the SEM.

switch was observed after treatment of A431 cells with the Hsp90 inhibitor geldanamycin (GA) (*Figure 6F*; *Ahsan et al., 2012*; *Gano and Simon, 2010*).

Since EGFR signaling induces the remodeling of the actin cytoskeleton in migratory cells (*Appert-Collin et al., 2015*), we tested if the absence of filopodia in A431 KO cells might be due to reduced EGFR signaling. Serum starvation of A431 control cells caused a strong reduction in filopodia formation (*Figure 6G*). Stimulation of serum starved A431 control cells with recombinant EGF induced the reappearance of actin-rich filopodia within 10 min (*Figure 6G'*). By contrast, EGF stimulation of serum starved A431 KO cells did not induce filopodia formation (*Figure 6H,H'*).

To directly measure the activity of the EGFR/RAS/MAPK pathway, we quantified ERK1/2 activity after EGF stimulation of serum-starved cells using a phospho-ERK1/2 specific antibody to probe western blots of total cell lysates (*Gabay et al., 1997*). In A431 control cells, phospho-ERK1/2 levels reached the maximum levels 5 min after EGF stimulation and declined almost to baseline levels within 30 min (*Figure 6I*). The total ERK1/2 levels did not change during the EGF stimulation. By contrast, phospho-ERK1/2 levels in A431 KO cells increased to around half of the maximal levels observed in A431 KO cells and decreased more rapidly.

Taken together, our results show that the human CHP-1 homolog CHORDC1 is required for EGF-induced filopodia formation and sustained ERK1/2 activation in A431 cells. Analogous to the results obtained for *C. elegans chp-1*, loss of CHORDC1 function does not eliminate but rather attenuates EGFR signaling in A431 cells.

## CHORDC1 controls the subcellular localization and stability of the EGFR

Since *chp-1* is required for the membrane localization of the LET-23 EGFR in *C. elegans*, we investigated if CHORDC1 also regulates EGFR localization in A431 cells. We analyzed receptor localization by immunofluorescence staining of fixed cells with an antibody against the extracellular domain of the EGFR. In A431 control cells, most of the EGFR signal was detected together with the actin-rich filopodia at the cell cortex, whereas only a small amount of EGFR staining was detected inside the cells (*Figure 7A*). In A431 KO cells, on the other hand, most of the EGFR staining was observed in intracellular punctae (*Figure 7B*). Orthogonal xz-projections through the cells revealed that most of the EGFR staining in A431 control cells overlapped with the cortical actin signal, while in A431 KO cells part of the signal was detected inside the cells and a fraction near the cortex underneath the cortical actin (*Figure 7A'–B''*). Moreover, A431 KO cells appeared significantly flatter than A431 control cells.

To examine if the loss of CHORDC1 results in a similar mislocalization of the EGFR to the ER as in *C. elegans chp-1(lf)* mutants, we co-stained the cells with antibodies against the EGFR and the ER marker PDI (protein disulfide isomerase) (*Jaronen et al., 2013*). While in A431 control cells only 5% of the EGFR signal colocalized with the PDI marker, 51% of the EGFR signal in the A431 KO cells overlapped with the PDI staining (*Figure 7C–E*).

Since CHORDC1 has been reported to act as a co-chaperone (*Gano and Simon, 2010*), we hypothesized that loss of CHORDC1 function might result in the incorrect folding of the EGFR and thereby cause its accumulation in the ER. To test this hypothesis, we performed trypsin sensitivity assays of the EGFR in A431 control versus KO cells. Since misfolded proteins are usually more susceptible to trypsin degradation, this assay can detect changes in protein folding (*Ninagawa et al., 2015*). Total protein extracts of A431 control and KO cells were incubated with varying concentrations of trypsin, and the amount of full-length EGFR was quantified by western blotting. As internal control, the EGFR signal intensities were normalized to the tubulin levels in the same lysates, as tubulin was degraded at approximately the same rate in the two cell populations. This assay revealed a faster degradation and thus an increased trypsin sensitivity of the EGFR in A431 KO cells (*Figure 7F*).

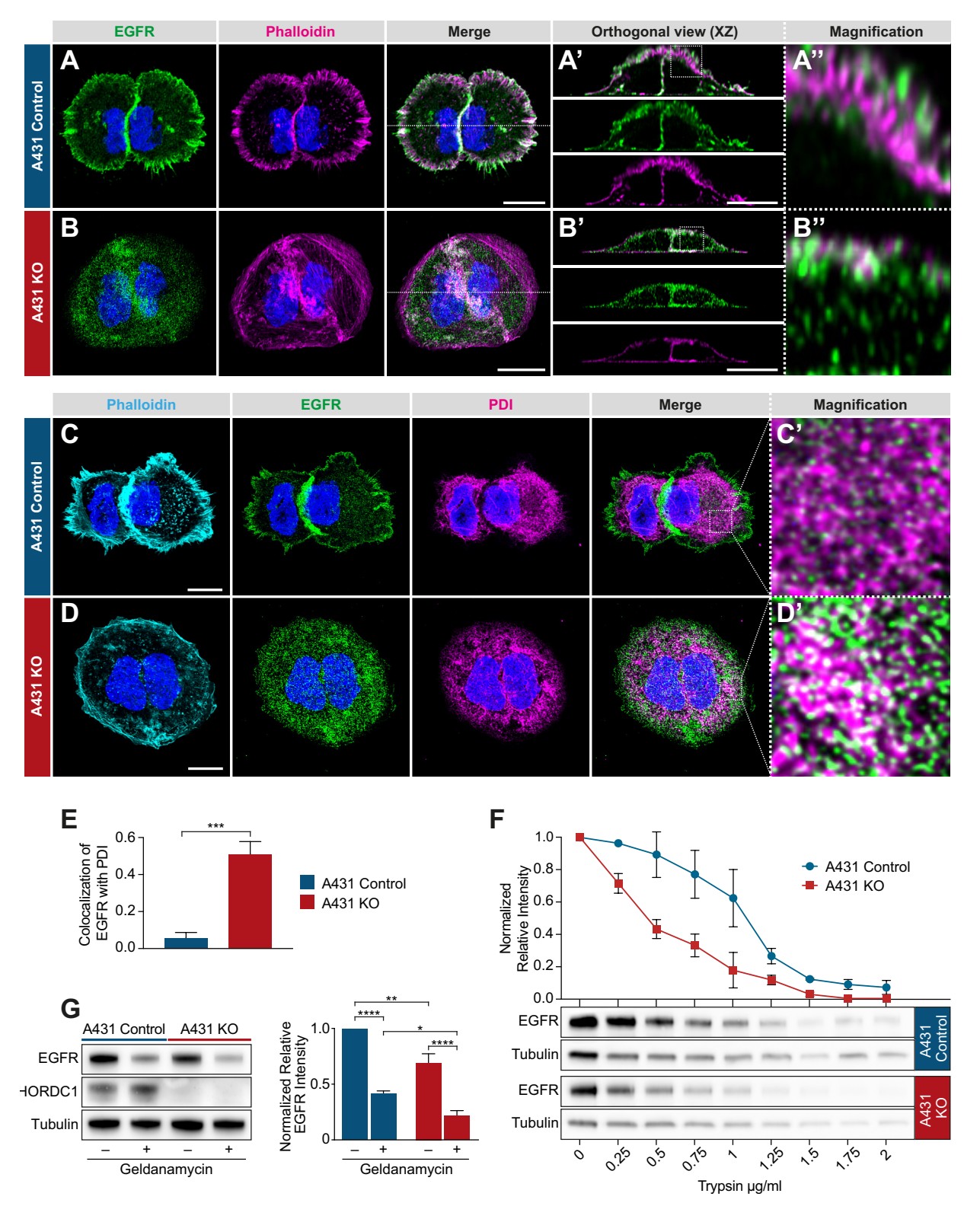

**Figure 7.** Mislocalization of the EGFR in CHORDC1 mutant A431 cells. (**A**) Immunofluorescence staining of A431 control and (**B**) A431 KO cells with antibodies recognizing EGFR (green), fluorescently labeled phalloidin (magenta) and DAPI (blue). The dotted lines in the merge panels indicate the planes used to create the orthogonal (XZ) views shown in (**A′, B′**). The doted squares in (**A′, B′**) indicate the areas shown at higher magnification in (**A′′, B′′**). (**C**) Immunofluorescence staining of A431 control- and (**D**) KO cells with fluorescently labeled phalloidin (light blue), antibodies recognizing EGFR

*Figure 7 continued on next page*

*Figure 7 continued*

(green), the endoplasmic reticulum marker PDI (magenta), and with DAPI (blue). (C', D') show higher magnifications of the regions outlined with dotted squares in the merge panels of (C, D). All images are maximum intensity projection of three confocal sections. The scale bars are 10 µm. (E) Co-localization of EGFR and PDI in A431 control (n = 5) and A431 KO cells (n = 10) was quantified by calculating the Mander's coefficient as described in Materials and methods (*Manders et al., 1993*). p-Values were calculated by a two-tailed t-test for independent samples. Error bars show the SEM. (F) Trypsin sensitivity assay. Total protein extracts of A431 control and A431 KO cells were incubated with the indicated trypsin concentrations, and the samples were analyzed by western blotting with antibodies against EGFR and tubulin. The line graph shows a quantification of the EGFR levels double normalized to the tubulin signal in each sample and to the undigested (0 µg/ml) samples. Error bars show the SEM. The average of three biological replicates is shown. (G) Western blot analysis of EGFR and CHORDC1 protein levels in A431 control and A431 KO cells with and without 1 µM geldanamycin treatment. The bar graph shows the normalized averaged relative intensities ± SEM. p-values ($p < 0.05$ = *, $p < 0.01$ = ** and $p < 0.0001$ = ***) were calculated by one-way ANOVA and corrected with a Tukey multiple comparison test. The average of three biological replicates is shown.

Consistent with earlier reports (*Ahsan et al., 2012*), the inhibition of HSP90 by geldanamycin caused an approximately two-fold reduction in total EGFR levels, while loss of CHORDC1 in A431 KO cells reduced EGFR levels only by 20% (*Figure 7G*). Notably, geldanamycin treatment further reduced the EGFR levels in A431 KO cells, below those detected in geldanamycin-treated A431 control cells. Thus, HSP90 stabilizes the EGFR at least in part independently of CHORDC1.

Taken together, we have found that the EGFR accumulates in the ER of A431 cells lacking CHORDC1, analogous to the mislocalization of LET-23 in *C. elegans chp-1(lf)* mutants. The increased trypsin sensitivity of the mislocalized EGFR in CHORDC1 mutant cells could be due to incorrect protein folding.

## Discussion

Intercellular signal transduction is regulated by the production and secretion of growth factors in the signal sending cells, as well as by the subcellular localization and intracellular trafficking of their receptors in the signal receiving cells (*Sorkin and Goh, 2009*). We have used vulval development in *C. elegans* as an in vivo model to study the the secretion and localization of the EGFR homolog LET-23. The basolateral localization and retention of LET-23 is essential for the efficient activation of the downstream RAS/MAPK pathway and correct VPC differentiation (*Whitfield et al., 1999*). Perturbations in LET-23 secretion or localization invariably cause defects in VPC fate specification and abnormal vulval morphogenesis.

In a screen for genes regulating LET-23 localization, we have previously identified the CHORD-containing protein CHP-1 as a regulator of LET-23 EGFR trafficking in the VPCs (*Haag et al., 2014*). Here, we show that the loss of *chp-1* function in *C. elegans* leads to the accumulation of LET-23 in the ER and a strong reduction -but not a complete inactivation- of RAS/MAPK signaling in the VPCs. Even in the absence of CHP-1, a small fraction of LET-23 reaches the plasma membrane, where it can bind to and be activated by the limiting amounts of LIN-3 EGF. Our genetic analysis confirms the notion that vulval induction in *chp-1(lf)* mutants depends to a large extent on *lin-3* activity. However, the ER accumulation of LET-23 in *chp-1(lf)* mutants is independent of *lin-3* or the basolateral *lin-2/lin-7/lin-10* receptor localization complex. This indicates that CHP-1 controls LET-23 secretion in the VPCs at an earlier step, before the receptor interacts with LIN-7 at the plasma membrane and undergoes LIN-3-mediated endocytosis.

Several observations have indicated that the role of *C. elegans* CHP-1 in LET-23 EGFR trafficking is specific. First, if CHP-1 was required for the ER trafficking of a large number of proteins, this would result in ER stress and activate the UPR pathway (*Calfon et al., 2002*). Yet, *chp-1(lf)* mutants did not exhibit an increased activity of the UPR pathway, unless additional ER stress was induced by globally inhibiting protein glycosylation. Second, the membrane localization of three other type I trans-membrane receptors, we examined (LIN-12 NOTCH, PAT-3 ß-integrin and LIN-18 RYK) did not change in *chp-1(lf)* mutants. Third, inhibition of other known HSP90 co-chaperones, such as *cdc-37*, *daf-41* and *sgt-1*, did not affect LET-23 localization in the VPCs.

In order to test if the function of CHP-1 in the EGFR signaling pathway is conserved in mammals, we inactivated the CHP-1 homolog CHORDC1 in human A431 epidermoid carcinoma cells, which express high levels of the wild-type EGFR and undergo a phenotypic switch in response to EGF

stimulation (*Ferretti et al., 2011*; *Van de Vijver et al., 1991*). The phenotype of the CHORDC1 knock-out in A431 cells is remarkably similar to the *chp-1(lf)* phenotype in the *C. elegans* VPCs; the EGFR accumulates in the ER and the activation of the RAS/MAPK pathway in response to EGF stimulation is strongly reduced. Even though the total levels of the EGFR are only slightly reduced in CHORDC1 mutant A431 cells, the EGFR is less stable as it exhibits an increased sensitivity to trypsin digestion (*Ninagawa et al., 2015*). The extracellular domain of the EGFR is N-glycosylated at multiple sites after entry into the ER lumen (*Azimzadeh Irani et al., 2017*). Since we did not observe a shift in the electrophoretic mobility of the EGFR in A431 KO cells, which would indicate a loss of glycosylation, it appears that the EGFR does enter the ER without CHORDC1. Possibly, CHORDC1 is required for the correct folding of the EGFR once it has entered the ER, and the partially unfolded EGFR molecules cannot pass through the ER.

It has been proposed that CHORDC1 acts as a co-chaperone that assists HSP90 in the folding of its numerous client proteins (*Gano and Simon, 2010*). According to this model, CHORDC1 would confer the specificity of HSP90 toward a subset of its clients, among them the EGFR. Surprisingly, our data in *C. elegans* point to an HSP90-independent function of CHP-1 in regulating EGFR trafficking. A loss-of-function mutation in the *C. elegans* HSP90 ortholog did not perturb LET-23 membrane localization, even though the total expression levels of LET-23 in the VPCs were reduced. By contrast, the VPCs in *chp-1(lf)* mutants did not exhibit an obvious reduction in LET-23 expression levels. Thus, HSP-90 likely stabilizes the LET-23 EGFR only after it has reached the VPC plasma membrane. We further tested if the HSP-90 paralog ENPL-1 might act together with CHP-1 instead of HSP-90, as the mammalian ENPL-1 homolog HSP90B1/GRP94 is required for the trafficking of certain transmembrane proteins though the ER (*Randow and Seed, 2001*). However, using an *enpl-1* deletion allele, we have excluded a possible involvement of ENPL-1 in LET-23 EGFR trafficking.

Even though our data on human cells cannot rule out the possibility that HSP90 and CHORDC1 might also act together, they are consistent with the HSP90-independent function of CHP-1 we observed in *C. elegans*. Geldanamycin inhibition of HSP90 in CHORDC1-deficient cells caused a significantly stronger reduction in EGFR levels than Geldanamycin treatment of wild-type cells, suggesting that HSP90 and CHORDC1 functions overlap only partially. One possible scenario is that a CDC37 co-chaperone/HSP90 chaperone complex stabilizes the EGFR at the plasma membrane (*Verba and Agard, 2017*), whereas CHORDC1 promotes the maturation and trafficking of EGFR through the ER. Along these lines, several co-chaperones were found to act independently of a core chaperone (*Echtenkamp and Freeman, 2012*). For example, p23, which contains the same ACD (alpha-crystallin-Hsps-p23-like) domain as CHORDC1, regulates various cellular processes that are distinct from those controlled by HSP90 (*Echtenkamp et al., 2011*). It is therefore possible that CHORDC1 acts by itself or in a complex with another, yet to be identified chaperone.

In summary, the identification of CHP-1/CHORDC1 as a specific regulator of EGFR trafficking extends an already complex network of different chaperones and co-chaperones that control receptor trafficking at different steps. Future studies may identify additional CHP-1/CHORDC1 membrane receptors that exhibit a similar mode of regulation.

## Materials and methods

### General *C. elegans* methods and strains

Unless specified otherwise, *C. elegans* strains were maintained at 20°C on Nematode Growth Medium (NGM) agar plates as described (*Brenner, 1974*). The *C. elegans* Bristol N2 strain was used as wild-type reference, and all strains generated through genetic crosses were derived from N2. A complete list of the *C. elegans* strains used can be found in the *Supplementary file 1*.

### RNAi feeding method

RNAi feeding experiments were performed as described previously (*Kamath and Ahringer, 2003*). The strain of interest was fed with *E. coli* HT115 expressing dsRNA against a specific target mRNA. Twenty synchronized L1 larvae were transferred to NGM plates containing 3 mM IPTG and 50 ng/ml ampicillin seeded with the indicated RNAi bacteria. The F1 progeny of the 20 P0 animals was analyzed at the L3 stage to score LET-23::GFP localization or at the L4 stage to examine vulval induction.

## Vulval induction

Vulval induction was scored by examining 20–40 worms of the indicated genotypes at the L4 stage under Nomarski optics. Animals were mounted on 4% agarose pads and anesthetized with 20 mM tetramisole in M9 buffer as described (*Sternberg and Horvitz, 1986*; *Schmid et al., 2015*). The vulval induction index (VI) was scored by counting the induced VPCs in 20–40 animals and calculating the average number of induced VPCs per animal. Statistical analysis is described in the legend to *Figure 4*.

## Tunicamycin treatment

The tunicamycin treatment to induce ER stress was carried out as described in *Taylor and Dillin (2013)*. Briefly, animals expressing the *hsp-4::gfp* reporter were synchronized at the L1 stage and allowed to develop on NGM plates until the first day of adulthood. Then, they were incubated for 4 hr at room temperature in 25 µg/ml tunicamycin solution in M9 buffer. Control animals were incubated in an equivalent dilution of DMSO, which was used as a solvent for tunicamycin. After the treatment, HSP-4::GFP expression was observed with a 10x lens on a Leica DM RA wide-field microscope. Images were analyzed using Fiji software (*Schindelin et al., 2012*), and the average intensity of the whole body in each animal was measured to make the box plot in *Figure 3—figure supplement 2*.

## Generation of the SP12 ER and AMAN-2 Golgi reporters

Plasmid constructs were made using Gibson Assembly cloning (*Gibson et al., 2009*) and verified by DNA sequencing. A list of the oligonucleotide primers used for plasmid construction can be found in the *Supplementary file 2*. To construct plasmid pAHE3 (P*dlg-1::aman-2::mCherry::unc-54 3'UTR*, the plasmid the pCFJ151 backbone (*Frøkjær-Jensen et al., 2008*) was recombined with the P*dlg-1* promoter and the *unc-54* 3' UTR, amplified as two individual fragments using the primers OEH153 and OEH158 and OEH159 and OEH156. The *aman-2* (F58H1.1) genomic sequence encoding the first 82 amino acids, including the signal sequence and transmembrane anchor, amplified from genomic DNA using the primers OEH152 and OEH155, and the mCherry coding sequence, amplified with the primers OEH154 and OEH157, were then inserted after the P*dlg-1* promoter.

To construct pAHE6 (P*dlg-1::mCherry::C34B2.10(SP12)::unc-54 3'UTR)*, the pCFJ151 backbone (*Frøkjær-Jensen et al., 2008*) was recombined with the P*dlg-1* promoter and the *unc-54* 3' UTR, amplified as two individual fragments using the primers OAHE22 and OEH159 and OEH158 and OEH153. The mCherry coding sequence, amplified with the primers OAHE19 and OAHE20, and the genomic sequence of C34B2.10 (SP12) containing the stop codon, amplified with the primers OAHE21 and OAHE8, were then inserted after the P*dlg-1* promoter. The primer OAHE21 contained an additional linker sequence encoding three Alanines for the N-terminal fusion with mCherry. For each of the two reporter plasmids, single copy insertion transgenes were generated by the MosSCI method as described (*Frøkjær-Jensen et al., 2008*).

## VPC-specific *chp-1* CRISPR/CAS9

Two target sites in the first exon of the genomic *chp-1* locus (sgRNA #1: CAG TGC TAT CAT AAA GGA TG and sgRNA #2 CGG TCT CCT TTT CGA TCC CA) were identified using the CRISPR design tool (http://crispr.mit.edu/) (*Hsu et al., 2013*). Double-stranded oligonucleotides were synthetized and cloned into the pDD162 (Addgene) to produce pAHE4 (sgRNA #1) and pAH5 (sgRNA #2). Plasmid pEV5 (*Pegl-17-Δpes-10::cas9*) (gift by Evelyn Lattmann) was used for 1° VPC-specific expression of the CAS9 protein. The plasmids pAHE4 and pAHE5 were co-injected into the gonads of wild-type animals as described (*Mello et al., 1991*) at a concentration of 50 ng/µl each together with the plasmid pEV5 at 100 ng/µl and the transformation markers pGH8 (P*rab-3::mCherry*) at 10 ng/ µl, pCFJ104 (P*myo-3::mCherry*) at 5 ng/µl and the pCFJ90 (P*myo-2::mCherry*) at 2.5 ng/µl to create the extrachromosomal array *zhEx558*.

## Mammalian cell culture

The human vulva epidermoid carcinoma cell line A431 was obtained from Sigma Aldrich (85090402), tested to be free of mycoplasm and cultured in Dulbecco's Modified Eagle Medium (Gibco 41966–

029) according to standard mammalian tissue culture protocols and sterile technique. DMEM was supplemented with 10% FCS (Gibco 10500–064) and 1% Pen-Strep (Gibco 15140–122).

## CHORDC1 knock-out in A431 cells

CHORDC1 guide RNAs targeting the first exon of CHORDC1 (CHORDC1 sgRNA #1: TTA CCG TCG GAA TTG GTC TC and CHORDC1 sgRNA #2: AGA CCA ATT CCG ACG GTA AG), as well as the scramble sequence GCA CTA CCA GAG CTA ACT CA, were identified using the CRISPR Design Tool (http://crispr.mit.edu/) (*Hsu et al., 2013*). Double-stranded oligos were generated and cloned into the lentiCRISPRv2 vector (Addgene), which was then transfected in combination with pVSV-G, pMDL and pREV into HEK293T cells to produce lentiviral particles (vMW6_CHORDC1 sg#1, vMW7_CHORDC1 sg#2, vMW9_scramble). Four days following transfection, the media from cells was collected, clarified by centrifugation, and filtered through a 0.45 µM filter to collect lentiviral particles. Subsequently, the particles were concentrated in Amicon Ultra tubes (Ultracel 100 k, Millipore). The titer of viral particles was determined before vMW6_CHORDC1 sg#1 and vMW7_CHORDC1 sg#2 were used to transduce 180'000 A431 cells in a 12-well plate at a combined MOI of 10. A431 cells were supplemented with DMEM media containing the lentiviral mix and 10 µg/ml polybrene and cultured under normal conditions. In parallel, cells were transduced with vMW9_scramble at a MOI of 10. Three days after transduction, cells were grown in the presence of 1.2 µg/ml puromycin. One week after puromycin selection, the puromycin-resistant populations were frozen and kept as stocks used in the subsequent experiments.

## Western blotting

For western blot analysis, cells were lysed in lysis buffer on ice (100 mM Tris/HCl, 150 mM NaCl, 1% Triton X, 1 mM EDTA, 1 mM DTT, 10 ml lysis buffer + 1 tablet protease inhibitor), scraped with a cell scraper and snap frozen in liquid nitrogen. 100 µl of this mix was sonicated in a Bioruptor sonicator device (Diagenode), before 100 µl of 2x SDS loading dye were added. About 10 µg of protein extract were resolved by SDS-PAGE and transferred to nitrocellulose membrane. The membrane was blocked in 5% dried milk in 1x PBS plus 0.2% Tween 20 and then incubated with the diluted primary antibodies overnight at 4°C. Secondary anti-rabbit or anti-mouse IgG antibodies conjugated to horseradish peroxidase (HRP) were used as the secondary antibodies. The HRP was detected by incubating the membrane with the SuperSignal West Pico or Dura Chemiluminescent Substrate (Thermo Scientific) for 4 min, before the signals were measured on a digital western blot imaging system. The antibodies used for Western blot analysis were: anti-CHORDC1 (HPA041040 Atlas Antibodies), anti-Tubulin (ab18251 abcam), anti-EGFR (HPA018530 Atlas Antibodies), anti-ERK1/2 (M5670 Sigma Aldrich), anti-ERK1/2 activated (M8159, Sigma Aldrich), HRP anti-Rabbit (111-035-144 Jackson Immuno Research) and HRP anti-Rabbit (115-035-146 Jackson Immuno Research). Quantification of western blots was done by measuring the band intensities in Fiji.

## Immunofluorescence staining of A431 cells

Cells were grown on glass slides in 24-well plates under standard conditions for 48 hr. Slides were then rinsed in PBS and fixed for 15 min in 4% PFA at 37°C. After washing with PBS, cells were permeabilized with PBS containing 0.2% Triton X-100% and 0.5% BSA for 5 min, and then blocked for 1 hr in PBS containing 0.5% BSA and 0.2% gelatine. Primary antibodies were added in blocking solution for 1 hr at room temperature in a humid chamber. The cells were rinsed in PBS three times before being incubated for 40 min in the dark with secondary antibodies and Phalloidin 568 (B3475 Thermo Scientific). After three washes with PBS, cells were stained for 5 min with PBS containing 0.1 µg/ml DAPI, followed by three washes with PBS. Glass slides were mounted with ProLong Gold Antifade Mountant (Thermo Scientific). The antibodies used for immunocytochemistry were: anti-CHORDC1 (HPA041040 Atlas Antibodies), anti-Tubulin (ab18251 abcam), anti-EGFR (MA5-13269 Thermo Scientific), anti-PDI (MA3-019 Thermo Scientific), Alexa Fluor 488 (A11034 Thermo Scientific), and Alexa Fluor 647 (A21236 Thermo Scientific). For all antibody stainings, at least three biological replicates were made.

## EGF stimulation

250'000 A431 cells were grown in 12-well plates under standard conditions for 24 hr. Thereafter, growth medium was replaced with DMEM lacking FCS. After 15 hr of starvation, cells were stimulated with 100 ng/ml human EGF (E9644 Sigma) for 10 min at 37°C, before they were lysed and prepared for western blot analysis.

## Trypsin sensitivity assay

750'000 A431 cells were seeded into a 25 cm$^2$ flask and grown until they reached confluency. After washing twice with cold PBS, cells were scraped with a cell scraper, lysed in lysis buffer (100 mM Tris pH = 8, 1% NP-40, 150 mM NaCl, 1 mM DTT) for 10 min on ice, sonicated for 10 min at 4°C, and clarified by centrifugation at 17,000 x g for 10 min at 4°C. Aliquots containing 50 µg of cleared protein samples were incubated with 0.1, 0.15, 0.2, 0.4, 0.6, 0.8, 1, 1.25 and 1.5 µg/ml Trypsin (Sigma EMS0004) and incubated for 15 min at 25°C while shaking. Thereafter, 2x SDS loading dye was added and samples were boiled prior to resolving the proteins by SDS-PAGE.

## Wide-field fluorescence microscopy

To examine the expression pattern of fluorescently tagged proteins, a Leica DM RA wide-field microscope equipped with a Hamamatsu ORCA-ER camera using a 40x/1.3 NA or 63x/1.4 NA oil immersion objective was used. Fluorescent and Nomarski mages were acquired with the Openlab 4.0 or VisiView 2.1 software. To observe the localization of GFP reporters, around 40 worms at the L3 stage were mounted on 4% agarose pads and anesthetized with 20 mM tetramisole in M9 buffer. In each experiment, the illumination intensity, camera exposure time and other software settings were kept constant.

## Confocal laser scanning microscopy

Around 40 larvae at the Pn.p to Pn.pxx stage were mounted on 4% agarose pads in M9 buffer containing 2 mM tetramisole. z-stacks at 0.2 to 0.3 µm spacing were taken using a Plan-Apochromat 63x/1.4 NA oil immersion objective on a Zeiss LSM710 confocal laser scanning microscope equipped with an 458/488/514 nm argon laser and a 594 nm helium-neon laser. Images were acquired by using the LSM710 ZEN 2012 software (Zeiss). GFP was excited with a 488 laser excitation and emission was detected in a range of 493–566 nm. mCherry was excited with a 594 nm laser excitation and emission was detected in a range of 599–696 nm. Images were captured with a variable frame size, a pinhole equivalent to 1 Airy and a pixel size of 0.08 µm. Identical camera gain settings were used for all live animal imaging (AMAN-2::mCherry: GFP 600, mCherry 500/mCherry::SP12: GFP 600, mCherry 600 in *Figure 3*). Images of antibody-stained A431 cells (*Figure 6* and *7*) were taken on a Leica CLSM SP8 upright microscope equipped with 405/488/552/638 nm solid state diode lasers. z-stacks at 0.16 µm spacing were taken using a 63x/1.4 HCX PL APO CS2 oil immersion objective with a variable frame size and a pinhole equivalent to 1 Airy.

## Image processing quantification of co-localization

Images were analyzed and processed with Fiji software (*Schindelin et al., 2012*) to adjust brightness and contrast. Images in *Figures 3*, *6* and *7* were deconvolved using the Huygens deconvolution software (Scientific Volume imaging). To quantify co-localization in *Figure 3* and *7*, raw images (before deconvolution) were processed in Fiji using the Subtract Background command (default settings, rolling ball radius = 0.3). To quantify co-localization, z-stacks were analyzed with the ImarisColoc module in Imaris 8.3. (Bitplane). The GFP channel of each image was thresholded to define a ROI using the mask channel function. An automatic threshold implemented in Imaris was set for both channels based on a statistical significance algorithm (*Costes et al., 2004*). With this approach, the extent of co-localization of two fluorescent-labeled proteins in an image is automatically quantified, without the bias of visual interpretation. Based on an automatically identified threshold calculated by the Coste's approach, a Manders Colocalization Coefficient (MCC) indicating the fraction of total probe fluorescence that co-localizes with the fluorescence of a second probe was calculated (*Manders et al., 1993*). The calculated MCC values were averaged, and statistical analysis was performed using a a two-tailed Student's t-test for independent samples.

## Acknowledgements

We wish to thank the members of the Hajnal laboratory for critical discussion and comments on the manuscript. We are also grateful to the *C. elegans* Genetics Center CGC, which is funded by NIH Office of Research Infrastructure Programs (P40 OD010440), the Mitani lab (National Bioresource Project) for providing some strains, Andrew Fire for GFP vectors, J Ahringer for RNAi clones and Franziska Walser and Marco Wachtel for their expertise in the production of lentiviral particles. Confocal imaging was performed with support of Urs Ziegler at the Center for Microscopy and Image Analysis, University of Zurich. This work was supported by a grant from the Swiss National Science Foundation to AH no. 31003A-166580 and the Kanton of Zürich.

## Additional information

### Funding

| Funder | Grant reference number | Author |
| --- | --- | --- |
| Swiss National Science Foundation | 31003A-166580 | Alex Hajnal |
| Kanton of Zürich | | Alex Hajnal |

The funders had no role in study design, data collection and interpretation, or the decision to submit the work for publication.

### Author contributions

Andrea Haag, Conceptualization, Formal analysis, Supervision, Validation, Investigation, Visualization, Methodology; Michael Walser, Adrian Henggeler, Conceptualization, Formal analysis, Validation, Investigation, Visualization, Methodology; Alex Hajnal, Conceptualization, Formal analysis, Supervision, Funding acquisition, Project administration

### Author ORCIDs

Michael Walser ⬦ http://orcid.org/0000-0003-3506-4621
Alex Hajnal ⬦ https://orcid.org/0000-0002-4098-3721

### Decision letter and Author response

Decision letter https://doi.org/10.7554/eLife.50986.sa1
Author response https://doi.org/10.7554/eLife.50986.sa2

## Additional files

### Supplementary files

- Supplementary file 1. Genotypes of the *C. elegans* strains used in this study.
- Supplementary file 2. Oligonucleotide primers used for plasmid constructions.
- Transparent reporting form

### Data availability

All data are included in the manuscript.

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
