## [Decision Letter]

**Acceptance summary:**

All reviewers found the identification of CHP-1 as a regulator of the trafficking and polarised expression of the EGF receptor has significant implications for the control of signalling. The apparent conservation of this function in mammalian cells also broadens the interest and appeal of this story. At a mechanistic level, there is much still to be understood, but the data imply that, unexpectedly, CHP-1 is acting independent of HSP90. This is significant because the mammalian homologue of CHP-1, CHORDC1, has been shown to be an HSP90 co-chaperone. This paper suggests that it may also have other molecular functions.

**Decision letter after peer review:**

[Editors’ note: the authors submitted for reconsideration following the decision after peer review. What follows is the decision letter after the first round of review.]

Thank you for submitting your work entitled “The CHORD protein CHP-1 regulates EGF receptor trafficking and signaling in *C. elegans* and human cells” for consideration by *eLife*. Your article has been reviewed by three peer reviewers, one of whom is a member of our Board of Reviewing Editors, and the evaluation has been overseen by a Senior Editor.

Our decision has been reached after consultation between the reviewers. Based on these discussions and the individual reviews below, we regret to inform you that your work will not be considered further for publication in *eLife*.

All three reviewers found the paper to be interesting and the subject matter potentially appropriate for *eLife*. There was also consensus that the authors have implicated CHP-1 in the biosynthetic pathway of the EGFR in worms. There were, however, significant concerns that preclude its publication in its current form.

The central concern was the conclusion that CHP-1 acts in an HSP90-independent manner. Neither of the two main strands of evidence for this were persuasive. The daf-21 allele is not a null, in fact being a weak gain of function, so the lack of EGFR accumulation in the ER does not address the HSP90 independence. A null allele would be needed to make this point. Similarly, as pointed out by two reviewers, the geldanamycin sensitivity of a population of CHORDC1 KO cells does not make a compelling case for HSP90 independence.

The reviewers do not believe that the paper can be published in *eLife* unless this issue were to be resolved rigorously. Moreover, if it turns out that CHP-1 is in fact HSP90-dependent, ie it is acting as expected, as a co-chaperone, the paper would then lack a degree of novelty to make a strong case for *eLife*. On those grounds – the substantial additional work needed to prove the relationship with HSP90 and the possibility that the result will undermine the novelty – the consensus decision was to decline the paper.

There were also other serious concerns, as well as a list of more minor issues. These are captured in the individual reviews below but the most significant include the following.

1) The lack of response of KO cells to serum does not provide strong evidence for an EGFR signalling defect.

2) The rab-5 knockdown experiments need positive controls to be convincing.

3) The genetic interactions between CHP-1 and other mutants in the let-23 pathway were inconclusive and not well explained.

4) The claims of medical significance were overstated.

Reviewer #1:

This paper describes the role of the co-chaperone CHP-1 in regulating the trafficking of the EGF receptor, both in *C. elegans* and in mammalian cells. The authors conclude that CHP-1 is required for the trafficking of the EGFR, although they do not provide a clear view of molecular mechanism. There is some evidence that EGFR is partially misfolded in CHP-1 mutants, supporting the idea that it's acting as an HSP90 co-chaperone; conversely, other evidence suggests it could be acting in an HSP90 independent manner.

Overall, this is a careful and interesting piece of work, introducing a new player in the biosynthetic/maturation pathway of the EGFR, a very important receptor in development, physiology and pathology. That argues for it being appropriate in principle for *eLife*. On the other hand, the story is not very extensive. The simple, solid conclusion is that CHP-1 is required in some rather poorly understood way to promote EGFR export from the ER.

The editorial decision in this case is therefore more about whether this represents a substantial enough advance for publication in *eLife*, than about major problems with the experiments. The specific concerns I have are all fairly easily addressable and don't, in my view, undermine the message substantially.

Specific issues

The mammalian cell data is interesting but there are a couple of weak elements. First, I don't agree that lack of response of mutant cells to serum demonstrates that they have direct defects in EGFR signalling. It is consistent with that interpretation, but the cells may have other reasons for being defective in reorganising their cytoskeleton. I do accept that the later result, assaying ERK phosphorylation, makes the case more directly.

My second concern about the mammalian data is the strong conclusion that CHP-1 acts independently of HSP90. Given that CHP-1 is known to be an HSP90 co-chaperone, a suggestion that it is acting in another way requires strong evidence. I am not persuaded that the sensitivity of CHP-1 KO cells to geldanamycin makes this case compellingly: the 'KO' cells actually have some residual activity (being a population rather than clones) and the drug may simply being inhibiting that residual level. Moreover, the trypsin sensitivity data does suggest that EGFR is significantly misfolded in the CHP-1 mutants, consistent with a co-chaperone role. The DAF21 data in worms also does not make a compelling case for HSP90 independence because they cannot assess the phenotype of a null allele, which is lethal. I do not argue that they are certainly wrong in their more radical model, but that they have not made a compelling case for it, yet they build it extensively into their conclusions and discussion.

The final paragraph of the Discussion, about possible therapeutic implications, is over-stated and inappropriate. Of course there are long-term possibilities that might stem from this kind of work, but the tone here is simply not justifiable.

A general point: much of the data are qualitative images but the conclusions depend on variable levels of staining, eg at the PM. Wherever practicable, it would be good to quantitate the results.

There is some slightly odd use of supplementary figures. I think that any data that contributes significantly to the main argument should be in primary figures. For example, I'd prefer the DAF21 data in supp 1A to be in the main figures because the authors argue quite forcefully about the CHP-1 role being HSP90 independent (an argument that I have explained that I do not find very convincing). Also Figure 3—figure supplement 2, about UPR induction. Isn't that quite central to their overall case?

The data in Figure 2D-F' is a bit confusing. They seem to imply that in the mutant background there is an increase (albeit modest) of EGFR in the Golgi and plasma membrane. Given their overall conclusion that CHP-1 promotes ER exit, how do they explain this?

The genetic interaction data are consistent with the model but don't add very much. The only surprising result, that loss of CHP-1 enhances a muv phenotype is not pursued, although a reasonable speculation is made that might explain it.

The requirement for CHP-1 in AC to P6p alignment is a perfectly plausible result but adds little to the overall message of the paper.

I don't understand their discussion about whether CHP-1 might be involved in entry of EGFR to the ER. Are they suggesting that it might be part of the co-translational insertion of the protein? I'd say that all their evidence implies that CHP-1 participates in (though is not absolutely essential for) onward trafficking from the ER. The only result that doesn't quite make sense is referred to above: the apparent increase of Golgi and PM EGFR in CHP-1 mutants.

Reviewer #2:

Haag et al. demonstrate a role for the HSP90 co-chaperone proteins CHP-1/CHORDC1 in the transit of the EGFR through the endoplasmic reticulum in both *C. elegans* and human cancer cells. Their data suggests that CHP-1/CHORDC1 functions independently of HSP90 to regulate EGFR trafficking. Loss of CHP-1/CHORDC1 results in reduced EGFR at the plasma membrane and reduced signaling through the Ras/MAPK pathway. In the *C. elegans* VPCs, CHP-1 appears to be a specific regulator of EGFR as the localization of other transmembrane proteins are not perturbed in *chp-1* mutants. Thus CHP-1/CHORDC1 may selectively regulate EGFR trafficking through the ER in humans and *C. elegans*. Overall the paper is very strong and there are no major scientific concerns. However, some statements do not align with the provided references (see below) and should be adjusted accordingly.

“LET-23 initially secreted to the basolateral membrane compartment, but after ligand-induced receptor endocytosis and transcytosis LET-23 accumulates on the apical cortex of the VPCs (Haag et al., 2014).”

We did not find sufficient evidence for this state in Haag et al., 2014. There was no data showing that LET-23 is secreted to the basolateral membrane prior to the apical. In fact, Whitfield et al., 1999, demonstrated LET-23 is initially apical using a LET-23 antibody. Haag et al., 2014 demonstrated that heat-shock expression of LIN-3 ligand increases the apical to basolateral ratio of LET-23 receptor. This is consistent with ligand-dependent endocytosis. Since it is not determined if the increased apical to basolateral ratio is due to only a decrease in basolateral localization versus a corresponding increase in apical localization, an argument for transcytosis cannot be made. In fact, there is no mention of transcytosis in Haag et al., 2014.

“The endocytosed receptor can be recycled to the basolateral compartment, transported by transcytosis to the apical membrane or lysosome degradation (Whitfield et al., 1999; Stetak et al., 2006).”

Whitfield et al. describes LET-23 localization by antibody staining, but does not mention trafficking.

Stetak et al., 2006 provide evidence that LET-23 is endocytosed, but no evidence of transcytosis of lysosome degradation.

Skorobogata and Rocheleau, 2012 (see below) show that RAB-7 and the ESCRT complex is required for endosome to lysosome trafficking would also support that LET-23 is endocytosed and trafficked to lysosomes.

“We performed RNAi against rab-5, which encodes a small GTPase that functions as a key regulator of early endosome formation and was previously shown to control LET-23 trafficking (Skorobogata and Rocheleau, 2012).”

There are no experiments testing rab-5 in this reference which instead focused on the rab-7 GTPase. Therefore, there is no previous evidence that rab-5 would prevent LET-23 endocytosis.

Reviewer #3:

Haag et al. show that the co-chaperone protein CHP-1 is required for normal plasma membrane localization of the LET-23 EGF receptor, but does not affect several other transmembrane proteins. In *chp-1* mutants, a large proportion of LET-23 colocalizes with the ER marker SP-12. LET-23 intracellular accumulation does not depend on ligand or endocytosis. A reporter of MAPK signaling shows reduced but not absent expression in P6.p daughter cells. Consistent with this, vulval differentiation occurs, though with abnormal morphology, and is ligand-dependent. Knocking out the human homologue CHORDC1 in A431 cells also reduces their responsiveness to EGF stimulation and causes EGFR mislocalization to the ER and increased sensitivity to trypsin.

The identification of EGFR as a CHORDC1 substrate and its HSP90-indepdendent function are potentially interesting. However, several points need to be addressed.

1) The HSP90 independence is not well demonstrated. The daf-21 allele used is a weak gain of function (Birnby et al., 2000), making its lack of LET-23 accumulation in the ER inconclusive. A null allele would have to be used to prove this point. The additional effect of geldanamycin on EGFR stability in CHORDC1 knockout cells does not prove HSP90 independence; HSP90 could act with another co-chaperone at an early step in EGFR folding, and with CHORDC1 at a later step that is required for ER export but not stability.

2) Showing that LET-23 mislocalization is independent of endocytosis relies on the lack of effect of rab-5 knockdown. A control demonstrating the effectiveness of this rab-5 RNAi in blocking endocytosis is needed.

3) The fact that vulval differentiation is only weakly affected, and is further reduced by loss of the ligand LIN-3, indicates that CHP-1 is not essential for LET-23 to reach the plasma membrane. Also, CHP-1 homologues have been shown to regulate other proteins such as Rho kinases. These observations weaken the authors' argument that CHORDC1 inhibitors would effectively and specifically inhibit EGFR signaling. They would need to more comprehensively assess which proteins are affected by loss of CHORDC1 in order to make the argument about its specificity.

---

## [Author Response]

[Editors’ note: The authors appealed the original decision. What follows is the authors’ response to the first round of review.]

The reviewers do not believe that the paper can be published in eLife unless this issue were to be resolved rigorously. Moreover, if it turns out that CHP-1 is in fact HSP90-dependent, ie it is acting as expected, as a co-chaperone, the paper would then lack a degree of novelty to make a strong case for eLife. On those grounds – the substantial additional work needed to prove the relationship with HSP90 and the possibility that the result will undermine the novelty – the consensus decision was to decline the paper.There were also other serious concerns, as well as a list of more minor issues. These are captured in the individual reviews below but the most significant include the following.1) The lack of response of KO cells to serum does not provide strong evidence for an EGFR signalling defect.2) The rab-5 knockdown experiments need positive controls to be convincing.3) The genetic interactions between CHP-1 and other mutants in the let-23 pathway were inconclusive and not well explained.4) The claims of medical significance were overstated.Reviewer #1:This paper describes the role of the co-chaperone CHP-1 in regulating the trafficking of the EGF receptor, both in *C. elegans* and in mammalian cells. The authors conclude that CHP-1 is required for the trafficking of the EGFR, although they do not provide a clear view of molecular mechanism. There is some evidence that EGFR is partially misfolded in CHP-1 mutants, supporting the idea that it's acting as an HSP90 co-chaperone; conversely, other evidence suggests it could be acting in an HSP90 independent manner.Overall, this is a careful and interesting piece of work, introducing a new player in the biosynthetic/maturation pathway of the EGFR, a very important receptor in development, physiology and pathology. That argues for it being appropriate in principle for eLife. On the other hand, the story is not very extensive. The simple, solid conclusion is that CHP-1 is required in some rather poorly understood way to promote EGFR export from the ER.The editorial decision in this case is therefore more about whether this represents a substantial enough advance for publication in eLife, than about major problems with the experiments. The specific concerns I have are all fairly easily addressable and don't, in my view, undermine the message substantially.Specific issuesThe mammalian cell data is interesting but there are a couple of weak elements. First, I don't agree that lack of response of mutant cells to serum demonstrates that they have direct defects in EGFR signalling. It is consistent with that interpretation, but the cells may have other reasons for being defective in reorganising their cytoskeleton. I do accept that the later result, assaying ERK phosphorylation, makes the case more directly.

This is a misunderstanding since we stimulated the serum starved cell with recombinant EGF in the absence of serum (subsection “*chp-1* is a positive regulator of EGFR/RAS/MAPK signaling in the VPCs”). Hence, the lacking response of the CHORDC1 mutant cells as well as the decreased ERK phosphorylation must be due to reduced EGFR activation.

My second concern about the mammalian data is the strong conclusion that CHP-1 acts independently of HSP90. Given that CHP-1 is known to be an HSP90 co-chaperone, a suggestion that it is acting in another way requires strong evidence. I am not persuaded that the sensitivity of CHP-1 KO cells to geldanamycin makes this case compellingly: the 'KO' cells actually have some residual activity (being a population rather than clones) and the drug may simply being inhibiting that residual level. Moreover, the trypsin sensitivity data does suggest that EGFR is significantly misfolded in the CHP-1 mutants, consistent with a co-chaperone role. The DAF21 data in worms also does not make a compelling case for HSP90 independence because they cannot assess the phenotype of a null allele, which is lethal. I do not argue that they are certainly wrong in their more radical model, but that they have not made a compelling case for it, yet they build it extensively into their conclusions and discussion.

The reviewer is right in her/his comment that the mammalian data by themselves do not prove an HSP90-independent function of CHP-1/CHORDC1. However, we were able to examine LET- 23::GFP localization in *hsp-90(ok1333)* loss-of-function worms at the Pn.p and Pn.px stage and found a similar effect as we originally described for the *hsp-90(p673)* allele (Figure 1 and Figure 1—figure supplement 1). We thus believe, the new *C. elegans* data clearly show an HSP-90-independent function of CHP-1 in the LET-23 pathway. The similarity of the phenotype in human CHORDC1 KO cells suggests a conserved function of CHORDC1 in EGFR trafficking, yet our mammalian data cannot exclude that CHORDC1 and HSP90 also function together. However, the Geldanamycin experiments indicate that HSP90 and CHORDC1 functions do not completely overlap. This is now discussed in subsection “Human CHORDC1 is required for filopodia formation and sustained ERK1/2 activation by the EGFR”.

Another important point, which we have addressed in the revised version, is the possibility that the HSP-90 paralog ENPL-1/HSP90B1 might act instead of HSP-90 together with CHP-1. However, we have ruled out this possibility by analyzing an *enpl-1* deletion mutant (Figure 1).

The final paragraph of the Discussion, about possible therapeutic implications, is over-stated and inappropriate. Of course there are long-term possibilities that might stem from this kind of work, but the tone here is simply not justifiable.

We have removed this entire paragraph from the Discussion to avoid any over-statements.

A general point: much of the data are qualitative images but the conclusions depend on variable levels of staining, eg at the PM. Wherever practicable, it would be good to quantitate the results.

Wherever possible, we have quantified the image data (e.g. Figures 3, 5, 7). However, we do not feel comfortable quantifying the intracellular LET-23::GFP signal intensity, e.g. in a *chp-1(lf)* mutant background since it is extremely difficult to draw a border between the cytoplasm and membrane in the relatively small VPCs. It should also be noted that the *chp-1(tm2277lf)* mislocalization phenotype is 100% penetrant (subsection “*chp-1* is required for basolateral localization of the EGFR LET-23”). Hence, the images shown in Figures 1, 2 and 4 are representative as there was little variation in the localization among animals of the same genotype- either LET-23::GFP was secreted to the plasma membrane or it accumulated entirely in the cytoplasm. For each genotype, we examined a minimum of 20 animals, and where we observed incomplete penetrance (e.g. Figure 2), we indicated this with a qualitative statement (i.e. xx out of yy animals showed the mislocalization phenotype), either in the figure legends (Figure 1 and 4) or in the figure (Figure 3). One exception was the *chp-1(lf); lin-2(lf)* phenotype, where a small fraction of the signal was seen apical in addition to the intracellular pool, as pointed out in subsection “Loss of *chp-1* function causes an accumulation of LET-23 EGFR in the endoplasmic reticulum”.

There is some slightly odd use of supplementary figures. I think that any data that contributes significantly to the main argument should be in primary figures. For example, I'd prefer the DAF21 data in supp 1A to be in the main figures because the authors argue quite forcefully about the CHP-1 role being HSP90 independent (an argument that I have explained that I do not find very convincing). Also Figure 3—figure supplement 2, about UPR induction. Isn't that quite central to their overall case?

The data with the *hsp-90* loss-of-function allele are now shown in the main Figure 1. The possibility of a URP in *chp-1* mutants is a relevant question that should be addressed. We therefore think the data merits being shown at least as figure supplement 2 to Figure 3.

The data in Figure 2D-F' is a bit confusing. They seem to imply that in the mutant background there is an increase (albeit modest) of EGFR in the Golgi and plasma membrane. Given their overall conclusion that CHP-1 promotes ER exit, how do they explain this?

Our other data -for example the *lin-3* sensitivity- indicate that a small fraction of LET-23::GFP does reach the plasma membrane, suggesting that the blockade in the ER is not absolute. It is therefore not surprising that a fraction of LET-23::GFP is found in the Golgi and that the Golgi localization is very slightly increased in *chp-1* mutants. See also our conclusion in subsection “*chp-1* acts cell autonomously in the VPCs”.

The genetic interaction data are consistent with the model but don't add very much. The only surprising result, that loss of CHP-1 enhances a muv phenotype is not pursued, although a reasonable speculation is made that might explain it.

Given the strong receptor mislocalization phenotype, we were surprised to see a very mild effect on the vulval induction index (VI) of *chp-1(lf)* single mutants. By examining double mutants with the *let-23* pathway, we could draw three main conclusions: (A) VPCs lacking CHP- 1 are still partially sensitive to the LIN-3 signal, (B) the output of the LET-23 pathway is decreased and (C) ligand sequestering is perturbed. We think, it is not only necessary to characterize the receptor mislocalization phenotype, but also investigate the physiological effects of the *chp-1(lf)* mutant on the output of the signaling pathway.

The requirement for CHP-1 in AC to P6p alignment is a perfectly plausible result but adds little to the overall message of the paper.

The AC alignment defects may not be surprising in light of the genetic interaction data, yet this phenotype is a second and independent piece of evidence for a partially reduced LET-23 activity in the VPCs. For this reason, we prefer to include these data in the manuscript.

I don't understand their discussion about whether CHP-1 might be involved in entry of EGFR to the ER. Are they suggesting that it might be part of the co-translational insertion of the protein? I'd say that all their evidence implies that CHP-1 participates in (though is not absolutely essential for) onward trafficking from the ER. The only result that doesn't quite make sense is referred to above: the apparent increase of Golgi and PM EGFR in CHP-1 mutants.

The reviewer is right in his comment that all our data indicate that the EGFR enters the ER, but its passage through the ER is blocked. We simply want to point to the extracellular glycosylation as another argument for ER entry of the EGFR in CHORDC1 mutant cells.

Reviewer #2:Haag et al. demonstrate a role for the HSP90 co-chaperone proteins CHP-1/CHORDC1 in the transit of the EGFR through the endoplasmic reticulum in both *C. elegans* and human cancer cells. Their data suggests that CHP-1/CHORDC1 functions independently of HSP90 to regulate EGFR trafficking. Loss of CHP-1/CHORDC1 results in reduced EGFR at the plasma membrane and reduced signaling through the Ras/MAPK pathway. In the C. elegans VPCs, CHP-1 appears to be a specific regulator of EGFR as the localization of other transmembrane proteins are not perturbed in chp-1 mutants. Thus CHP-1/CHORDC1 may selectively regulate EGFR trafficking through the ER in humans and C. elegans. Overall the paper is very strong and there are no major scientific concerns. However, some statements do not align with the provided references (see below) and should be adjusted accordingly.“LET-23 initially secreted to the basolateral membrane compartment, but after ligand-induced receptor endocytosis and transcytosis LET-23 accumulates on the apical cortex of the VPCs (Haag et al., 2014).”We did not find sufficient evidence for this state in Haag et al., 2014. There was no data showing that LET-23 is secreted to the basolateral membrane prior to the apical. In fact, Whitfield et al., 1999, demonstrated LET-23 is initially apical using a LET-23 antibody. Haag et al., 2014 demonstrated that heat-shock expression of LIN-3 ligand increases the apical to basolateral ratio of LET-23 receptor. This is consistent with ligand-dependent endocytosis. Since it is not determined if the increased apical to basolateral ratio is due to only a decrease in basolateral localization versus a corresponding increase in apical localization, an argument for transcytosis cannot be made. In fact, there is no mention of transcytosis in Haag et al., 2014.“The endocytosed receptor can be recycled to the basolateral compartment, transported by transcytosis to the apical membrane or lysosome degradation (Whitfield et al. 1999; Stetak et al., 2006).”Whitfield et al. describes LET-23 localization by antibody staining, but does not mention trafficking.Stetak et al., 2006 provide evidence that LET-23 is endocytosed, but no evidence of transcytosis of lysosome degradation.Skorobogata and Rocheleau, 2012 (see below) show that RAB-7 and the ESCRT complex is required for endosome to lysosome trafficking would also support that LET-23 is endocytosed and trafficked to lysosomes.

Regarding the point of transcytosis:

The reviewer is correct in that the cited references do not directly demonstrate transcytosis. This statement rather reflects our interpretation of the existing data and transcytosis of LET-23 has not been directly demonstrated in *C. elegans*. To be more conservative, we have removed the mention of transcytosis from the manuscript. We have also removed the Kaech et al. reference at this position since basolateral localization was only demonstrated later by Whitfield et al.

Regarding the point of lysosomal degradation:

Stetak et al. showed that blocking LET-23 endocytosis prevented the degradation of LET-23 in VPCs adjacent to P6.p, where LET-23 is initially expressed but disappears after induction. Skorobogata and Rocheleau showed that the pathway required for lysosomal trafficking controls LET-23. Lysosomal degradation of EGFRs in mammalian cells is well documented, and the existing *C. elegans* data suggest that lysosomal degradation of LET-23 in the VPCs does occur, especially in the 2° and 3° VPCs (see modifications and new reference in the text).

“We performed RNAi against rab-5, which encodes a small GTPase that functions as a key regulator of early endosome formation and was previously shown to control LET-23 trafficking (Skorobogata and Rocheleau, 2012).”There are no experiments testing rab-5 in this reference which instead focused on the rab-7 GTPase. Therefore, there is no previous evidence that rab-5 would prevent LET-23 endocytosis.

This comment is similar to a point raised by reviewer #3 below. Since the *rab-5* data are based on a negative RNAi result and *rab-5* mutants die as young larvae, the RNAi experiment may be inconclusive and mutant analysis was not possible. We removed the entire *rab-5* experiment from the revised manuscript as it is not an essential argument. The *chp-1(lf); lin-3(rf)* double mutant phenotype makes the important point that *chp-1* is not involved in ligand-induced receptor endocytosis (Figure 4).

Reviewer #3:Haag et al. show that the co-chaperone protein CHP-1 is required for normal plasma membrane localization of the LET-23 EGF receptor, but does not affect several other transmembrane proteins. In chp-1 mutants, a large proportion of LET-23 colocalizes with the ER marker SP-12. LET-23 intracellular accumulation does not depend on ligand or endocytosis. A reporter of MAPK signaling shows reduced but not absent expression in P6.p daughter cells. Consistent with this, vulval differentiation occurs, though with abnormal morphology, and is ligand-dependent. Knocking out the human homologue CHORDC1 in A431 cells also reduces their responsiveness to EGF stimulation and causes EGFR mislocalization to the ER and increased sensitivity to trypsin.The identification of EGFR as a CHORDC1 substrate and its HSP90-indepdendent function are potentially interesting. However, several points need to be addressed.1) The HSP90 independence is not well demonstrated. The daf-21 allele used is a weak gain of function (Birnby et al., 2000), making its lack of LET-23 accumulation in the ER inconclusive. A null allele would have to be used to prove this point.

This is the critical point, similar to the second argument made above by reviewer #1. We have addressed this issue by investigating LET-23::GFP localization in the *hsp-90(ok1333)* loss-of function background. Most homozygous *ok1333* larvae arrested in the mid- to late-L3 stage, allowing us to examine LET-23::GFP localization at the Pn.p and Pn.px stage (Figure 1). Since the *p673* allele has a similar effect as *ok1333* on LET-23::GFP localization (Figure 1—figure supplement 1), we propose that with respect to LET-23 localization, *p673* acts as a reduction-of-function rather than a gain-of-function allele. Moreover, we have excluded the possibility that the HSP-90 paralog ENPL-1/HSP90B1 might act instead of HSP-90 together with CHP-1 (Figure 1).

The additional effect of geldanamycin on EGFR stability in CHORDC1 knockout cells does not prove HSP90 independence; HSP90 could act with another co-chaperone at an early step in EGFR folding, and with CHORDC1 at a later step that is required for ER export but not stability.

The reviewer is correct with this argument and we explicitly state this point. The Geldanamycin experiment does not prove HSP90 independence, it just indicates that the functions of HSP90 and CHORDC1 do not completely overlap, which is consistent with the *C. elegans* data.

2) Showing that LET-23 mislocalization is independent of endocytosis relies on the lack of effect of rab-5 knockdown. A control demonstrating the effectiveness of this rab-5 RNAi in blocking endocytosis is needed.

We agree, a negative RNAi experiment is inconclusive and *rab-5* mutants are early larval lethal. As mentioned above in response to a similar comment made by reviewer #2, we have removed the *rab-5* experiment from the revised manuscript. The *chp-1(lf); lin-3(rf)* double mutant phenotype already makes the important point that *chp-1* is not involved in ligand-induced receptor endocytosis (Figure 4).

3) The fact that vulval differentiation is only weakly affected, and is further reduced by loss of the ligand LIN-3, indicates that CHP-1 is not essential for LET-23 to reach the plasma membrane. Also, CHP-1 homologues have been shown to regulate other proteins such as Rho kinases. These observations weaken the authors' argument that CHORDC1 inhibitors would effectively and specifically inhibit EGFR signaling. They would need to more comprehensively assess which proteins are affected by loss of CHORDC1 in order to make the argument about its specificity.

As pointed out in our response to a similar comment made by reviewer #1, we have removed this – admittedly- very speculative idea on the potential use of CHORDC1 inhibitors from the Discussion section. The conclusion that *chp-1(lf)* does not completely eliminate LET-23 function and ligand binding is mentioned in several places in the Results section and summarized in the Discussion. It is very likely that CHP-1/CHORDC1 has multiple targets in addition to the EGFR and ROCK. While the adult sterile phenotype of *chp-1(lf)* may very well be explained by altered ROCK activity, we have previously examined a possible role of *C. elegans* ROCK using a *let-502* deletion allele, but found no effect on LET-23::GFP localization (Haag and Escobar, unpublished results). With respect to other trans-membrane receptors, we are still screening candidates, but so far, have not found any besides LET-23.